# Molecular basis of outer kinetochore assembly on CENP-T

Pim J Huis in 't Veld[1*†], Sadasivam Jeganathan[1†‡], Arsen Petrovic[1], Priyanka Singh[1], Juliane John[1], Veronica Krenn[1§], Florian Weissmann[2,3], Tanja Bange[1], Andrea Musacchio[1,4,5*]

[1]Department of Mechanistic Cell Biology, Max Planck Institute of Molecular Physiology, Dortmund, Germany; [2]Research Institute of Molecular Pathology (IMP), Vienna, Austria; [3]Vienna Biocenter (VBC), Vienna, Austria; [4]Centre for Medical Biotechnology, University Duisburg-Essen, Essen, Germany; [5]Faculty of Biology, University Duisburg-Essen, Essen, Germany

*For correspondence: pim.huis@mpi-dortmund.mpg.de (PH); andrea.musacchio@mpi-dortmund.mpg.de (AM)

[†]These authors contributed equally to this work

Present address: [‡]Chemical Genomics Centre of the Max Planck Society, Dortmund, Germany; [§]Institute of Molecular Biotechnology of the Austrian Academy of Sciences (IMBA), Vienna Biocenter (VBC), Vienna, Austria

**Abstract** Stable kinetochore-microtubule attachment is essential for cell division. It requires recruitment of outer kinetochore microtubule binders by centromere proteins C and T (CENP-C and CENP-T). To study the molecular requirements of kinetochore formation, we reconstituted the binding of the MIS12 and NDC80 outer kinetochore subcomplexes to CENP-C and CENP-T. Whereas CENP-C recruits a single MIS12:NDC80 complex, we show here that CENP-T binds one MIS12:NDC80 and two NDC80 complexes upon phosphorylation by the mitotic CDK1:Cyclin B complex at three distinct CENP-T sites. Visualization of reconstituted complexes by electron microscopy supports this model. Binding of CENP-C and CENP-T to MIS12 is competitive, and therefore CENP-C and CENP-T act in parallel to recruit two MIS12 and up to four NDC80 complexes. Our observations provide a molecular explanation for the stoichiometry of kinetochore components and its cell cycle regulation, and highlight how outer kinetochore modules bridge distances of well over 100 nm.

## Introduction

Accurate chromosome segregation in eukaryotes requires the coordinated action of hundreds of proteins. Subsets of these assemble on centromeric chromatin that is epigenetically specified by the enrichment of centromeric protein A (CENP-A), a variant of Histone H3 (*Guse et al., 2011*). These assemblies, named kinetochores, form the major point of attachment between centromeres and the mitotic or meiotic spindle and couple the force of depolymerizing microtubules to chromosome movement (*Cheeseman, 2014*). Kinetochores also function as signaling platforms for the spindle assembly checkpoint and delay the onset of chromosome segregation in the presence of erroneous chromosome-spindle attachments (*Musacchio, 2015*).

The structural core of the kinetochore assembles from two large multi-subunit complexes, the CCAN (constitutive centromere associated network), and the KMN (KNL1 complex, MIS12 complex, NDC80 complex, abbreviated as KNL1C, MIS12C, and NDC80C), which are respectively proximal and distal to the centromeric chromatin (*Pesenti et al., 2016*). The affinity of the kinetochore for microtubules in the KMN is predominantly mediated by the NDC80 complex, a heterotetramer with an approximately 50 nm long coiled coil region that separates the microtubule-binding calponin homology (CH) domains of the NDC80 and NUF2 subunits from the SPC24 and SPC25 subunits (*Ciferri et al., 2005*; *Wei et al., 2005*, *2007*; *Ciferri et al., 2008*). NDC80C is essential to form stable regulated kinetochore-microtubule interactions and its localization at the outer kinetochore is thus a prerequisite for faithful chromosome segregation (*DeLuca et al.,*

*2005*, *2006*; *Cheeseman et al., 2006*). The recruitment of NDC80C requires the CCAN subunits CENP-C and CENP-T (*Liu et al., 2006*; *Okada et al., 2006*; *Kwon et al., 2007*; *Carroll et al., 2010*; *Gascoigne et al., 2011*; *Hori et al., 2013*) as well as a nuclear envelope breakdown and the activity of mitotic kinases (*Gascoigne and Cheeseman, 2013*).

Whereas CENP-C directly binds to CENP-A containing nucleosomes (*Carroll et al., 2010*) and has been proposed to act as a blueprint for further kinetochore assembly (*Klare et al., 2015*), CENP-T is integrated in a CENP-TWSX complex that requires its DNA-binding activity (*Hori et al., 2008*; *Nishino et al., 2012*), as well as interactions with the CCAN (*Carroll et al., 2010*; *Basilico et al., 2014*; *Logsdon et al., 2015*; *Samejima et al., 2015*; *Suzuki et al., 2015a*; *Pekgöz Altunkaya et al., 2016*), to localize to the kinetochore.

Depletions of CENP-C or CENP-T, or prevention of Cyclin-dependent kinase (CDK) phosphorylation of CENP-T all affect proper recruitment of MIS12C and NDC80C to the kinetochore (*Gascoigne et al., 2011*; *Gascoigne and Cheeseman, 2013*; *Hori et al., 2013*; *Kim and Yu, 2015*; *Rago et al., 2015*; *Suzuki et al., 2015a*). Studying the respective contribution of CENP-C and CENP-T to outer kinetochore assembly, however, has been complicated by their interdependent localization and their distinct regulation by post-translational modifications. Experiments that targeted either CENP-C or CENP-T to an ectopic chromatin array uncoupled these requirements and demonstrated the ability of both pathways to recruit outer kinetochore components (*Gascoigne et al., 2011*; *Hori et al., 2013*). An important conclusion from these previous studies is that CENP-C and CENP-T recruit NDC80C in distinct ways (*Figure 1A*). The CENP-C-dependent axis, which is now understood in detail, relies on a direct interaction of CENP-C with the MIS12 complex (MIS12C) (*Screpanti et al., 2011*; *Przewloka et al., 2011*; *Dimitrova et al., 2016*; *Petrovic et al., 2016*), which further recruits NDC80C via a direct interaction (*Cheeseman et al., 2006*; *Petrovic et al., 2010*).

The CENP-T-dependent pathway, on the other hand, remains less well characterized. It is firmly established that CENP-T interacts directly with NDC80C, with (humans and chicken) or without (budding yeast) prior phosphorylation by CDK1:Cyclin B of sequence motifs in the CENP-T N-terminal region (*Gascoigne et al., 2011*; *Schleiffer et al., 2012*; *Malvezzi et al., 2013*; *Nishino et al., 2013*; *Pekgöz Altunkaya et al., 2016*). Complicating the picture, however, CENP-T also contributes to kinetochore recruitment of MIS12C (*Kim and Yu, 2015*; *Rago et al., 2015*). This may appear surprising, because MIS12C and CENP-T bind to NDC80C in a competitive manner (*Schleiffer et al., 2012*; *Nishino et al., 2013*), suggesting that they can only recruit NDC80C independently from each other. Whether the interaction of CENP-T with MIS12C is direct, and compatible with CENP-T binding to NDC80C, is therefore currently unclear.

The ability of kinetochores to form multiple low-affinity linkages with microtubules is at the basis of popular models of kinetochore-microtubule attachment such as the Hill's sleeve (*Hill, 1985*). It is also plausible that crucial regulatory aspects of kinetochore-microtubule attachment, such as the tension-dependent regulation required for correcting erroneous attachments or for stabilization of correct ones, depend on the effective number and distribution of linkages, as recently proposed in elegant modeling studies (*Zaytsev et al., 2014*). A detailed understanding of the stoichiometry of outer kinetochore composition and of its regulation is therefore crucial.

To address this question, we have embarked on an effort of biochemical reconstitution of kinetochores. This recently allowed us to report that the CENP-A nucleosome recruits two copies of the CCAN complex (*Weir et al., 2016*). In this study, we address instead the composition and regulation of the interactions between CCAN and the outer kinetochore, focusing on the interactions of CENP-T, MIS12C, and NDC80C. CDK1:Cyclin B phosphorylation reactions with systematically mutated CENP-T substrates revealed that the phosphorylation of CENP-T at Thr11 and Thr85 results in the recruitment of two NDC80Cs. We also report that CENP-T interacts directly with MIS12C. This requires phosphorylation of CENP-T at Ser201, which is encompassed within a non-canonical CDK1 consensus sequence. This CENP-T-bound MIS12C recruits additional NDC80C and KNL1C complexes. Importantly, CENP-C and CENP-T bind MIS12C in a competitive manner and form separate outer kinetochore recruitment modules. Thus, we have reconstituted and visualized the molecular basis of CENP-T mediated assembly of the outer kinetochore and its phospho-regulation.

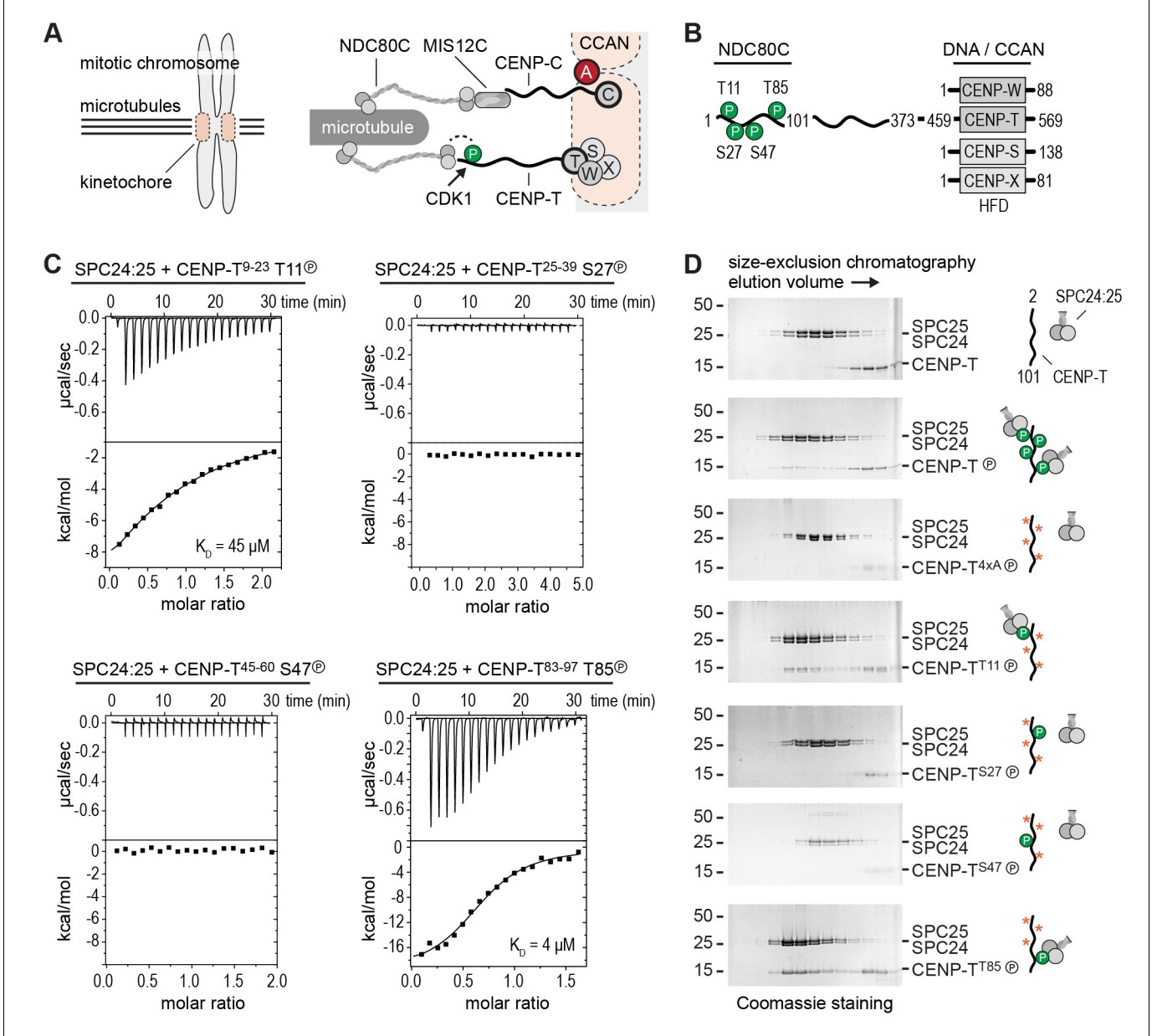

**Figure 1.** Phosphorylation of CENP-T$^{2-101}$ at T11 or T85 is sufficient for the binding of SPC24:SPC25. (A) Schematic representation of CENP-C and CENP-T recruiting MIS12C:NDC80C and NDC80C. (B) Besides the histone-fold domain (HFD) at the carboxy terminus, CENP-T mainly consists of regions of compositional bias, rich in polar residues, and likely to be intrinsically disordered. The boundaries of CENP-T constructs used in this study and domains involved in the binding of NDC80C and DNA/CCAN are indicated. In vitro phosphorylated residues in CENP-T$^{1-101}$ are marked by a P (see also *Table 1*). (C) The binding between SPC24:SPC25 and CENP-T phosphopeptides was determined by isothermal titration calorimetry. The y-axis indicates kcal/mole of injectant. Dissociation constants ($K_d$) between SPC24:SPC25 and phosphopeptides containing T11 and T85 were determined to be 45 µM and 4 µM respectively. (D) SDS-PAGE analysis of various CENP-T$^{2-101}$ mutants that were incubated with SPC24:SPC25 and separated by analytical size-exclusion chromatography. Red asterisks indicate mutated phosphorylation sites. See *Figure 1—figure supplement 1* for the entire dataset.

The following figure supplement is available for figure 1:

**Figure supplement 1.** Phosphorylation of CENP-T$^{2-101}$ at T11 or T85 is sufficient for the binding of SPC24:SPC25.

## Results

### Phosphorylation of CENP-T at Thr11 or Thr85 is sufficient to recruit SPC24:SPC25

To obtain insight into the molecular mechanism of CENP-T mediated recruitment of the human outer kinetochore, we set out to reconstitute this process using purified components. Previous studies identified multiple phosphorylation sites in CENP-T and showed that its phosphorylation contributes to the recruitment of NDC80C (*Gascoigne et al., 2011*; *Kettenbach et al., 2011*; *Gascoigne and Cheeseman, 2013*; *Nishino et al., 2013*; *Rago et al., 2015*). Confirming this, we identified by mass spectrometry numerous residues in CENP-T that were in vitro phosphorylated by CDK1:Cyclin B (*Table 1*). These included the CDK target sites Thr11, Thr27, Ser47, and Thr85 in the N-terminal region of CENP-T, a region known to interact with the SPC24:SPC25 subunits of NDC80C (*Malvezzi et al., 2013*; *Nishino et al., 2013*) (*Figure 1B*). We used isothermal titration calorimetry to test the ability of four short synthetic phosphopeptides encompassing these residues to bind SPC24:SPC25. Peptides containing phosphorylated Thr11 or Thr85 showed a binding affinity of 45 µM and 4 µM, respectively, whereas peptides containing phosphorylated Thr27 or Ser47 did not show an interaction with SPC24:SPC25 (*Figure 1C*). Although the determined affinities are modest and unlikely to fully represent the binding affinities between full-length CENP-T and SPC24:SPC25, they distinguish *bona fide* binding sites from sites that do not interact.

Confirming these equilibrium binding experiments, the interaction between an N-terminal fragment of purified CENP-T and SPC24:SPC25 depended on the phosphorylation of CENP-T by CDK1: Cyclin B (*Figure 1D* and *Figure 1—figure supplement 1*). We next generated a set of CENP-T[2-101] constructs in which three of the four CDK phosphorylation sites were mutated to Alanine. Constructs retaining either Thr11 or Thr85 bound to SPC24:SPC25 in a phosphorylation dependent manner, whereas constructs retaining either Thr27 or Ser47, or having all four sites mutated to Alanine, did not bind SPC24:SPC25 (*Figure 1D* and *Figure 1—figure supplement 1*). Phosphorylation of CENP-T at Thr11 or Thr85 is thus sufficient to bind SPC24:SPC25. These results are consistent with in vivo experiments showing that ectopic chromosome anchoring of a CENP-T[1-250] fragment results in the recruitment of NDC80C when either Thr11 or Thr85 were mutated to Alanine, but not when both were mutated (*Rago et al., 2015*).

**Table 1.** Phospho-sites on CENP-T[2-373] in vitro phosphorylated by CDK1:Cyclin B. Identified phospho-peptides with a localization probability for the phospho-site >80% and an Andromeda search engine score >100 are shown. All peptides have a posterior error probability (PEP) below $1 \times 10^{-25}$. The last column indicates if sites were identified as single (S) and/or double (D) phosphohorylated peptides.

| Position | Peptide with pS or pT | Score | P-sites |
|---|---|---|---|
| 8 | MADHNPD**pS**DS**pT**PRTLLR | 144 | S and D |
| 10 | MADHNPDSD**pS**TPRTLLR | 209 | S |
| 11 | MADHNPD**pS**D**pS**TPRTLLRR | 209 | S and D |
| 27 | VLDTADPR**pT**PR | 135 | S |
| 45 | RALLE**pT**A**pS**PRK | 112 | D |
| 47 | RALLE**pT**A**pS**PRK | 112 | S and D |
| 78 | SAHIQA**pS**GHLEEQ**pT**PR | 101 | S and D |
| 85 | SAHIQA**pS**GHLEEQ**pT**PR | 167 | S and D |
| 184 | LSVFQQGGVDQGLSLSQEPQGNADAS**pS**LTR | 129 | S |
| 195 | SLNLTFA**pT**PLQPQ**pS**VQRPGLAR | 152 | S and D |
| 201 | SLNLTFA**pT**PLQPQ**pS**VQRPGLAR | 186 | S and D |

## CENP-T coordinates two NDC80 complexes in a single complex

The finding that phosphorylation of CENP-T at either Thr11 or Thr85 is sufficient to recruit SPC24:SPC25, both in vivo (*Rago et al., 2015*) and in vitro (*Figure 1*), raises the question whether phosphorylated CENP-T can bind two SPC24:SPC25 units simultaneously. This possibility is supported by the change in retention volume and the apparent stoichiometry of the CENP-T$^{2-101}$:SPC24:SPC25 complex (*Figure 1D*). We set out to test the stoichiometry of CENP-T:SPC24:SPC25 complexes further using a CENP-T$^{2-373}$ fragment that we C-terminally labeled with a fluorescent dye to specifically follow CENP-T during chromatography and SDS-PAGE (*Figure 2*). This is helpful because CENP-T$^{2-373}$, which lacks Tryptophan and has only one Tyrosine, absorbs poorly at 280 nm. Fluorescently labeled CENP-T$^{2-373}$ (hereafter CENP-T) was in vitro phosphorylated by CDK1:Cyclin B and subsequently incubated with a threefold molar excess of SPC24:SPC25. CENP-T was part of a broad peak with two SPC24:SPC25-containing species with approximate retention volumes of 1.2 and 1.35 ml (*Figure 2*, green traces). Consistent with the idea that the complex eluting at 1.2 mL contained two SPC24:SPC25 molecules per CENP-T, this species was not observed for CENP-T containing the T11A or the T85A mutation (*Figure 2*, orange traces). CENP-T that was mutated at both Thr11 and

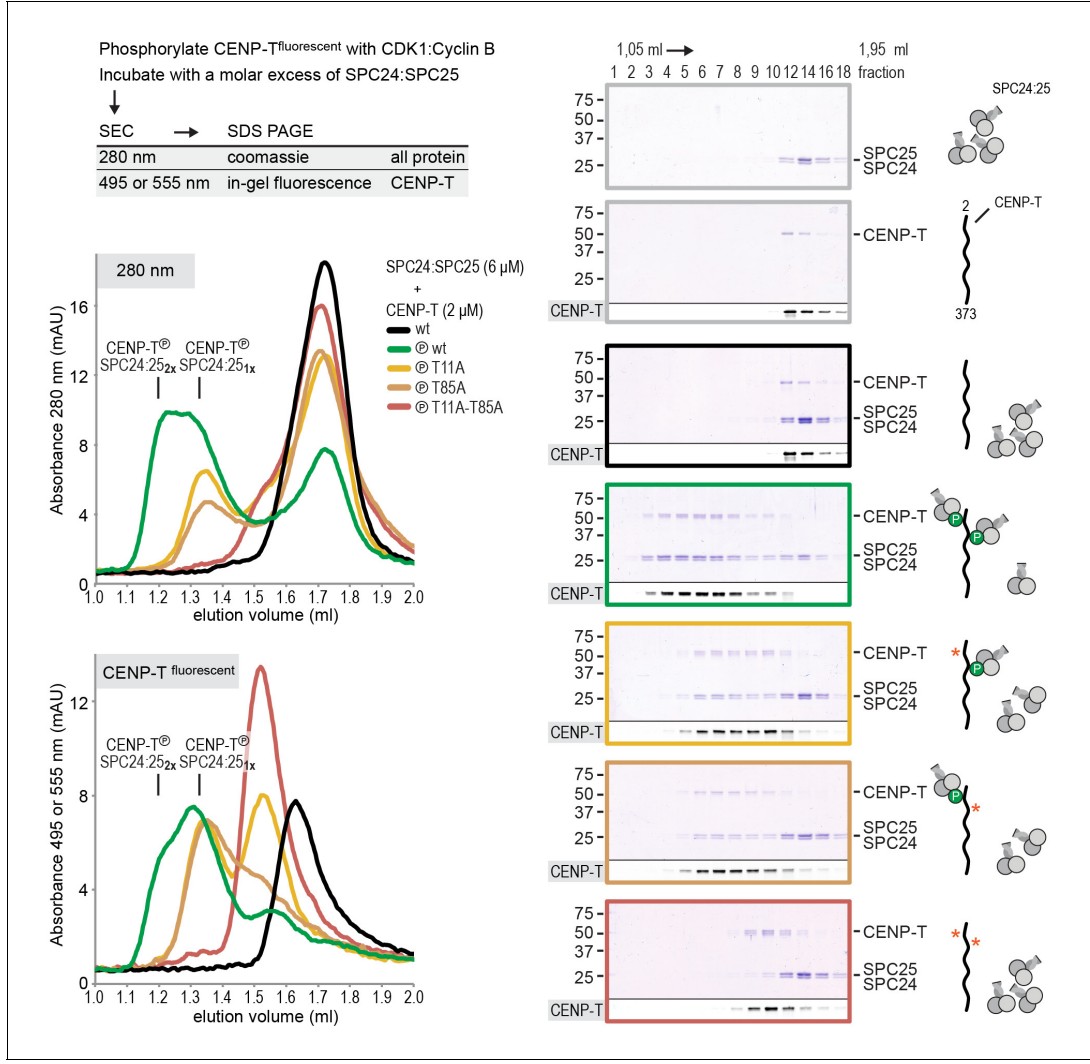

**Figure 2.** CENP-T phosphorylated at T11 and T85 binds two copies of SPC24:SPC25. Analytical size-exclusion chromatography (Superdex 200 5/150 increase) and SDS-PAGE show that CENP-T$^{2-373}$ phosphorylated by CDK1:Cyclin B at positions T11 and T85 can bind two copies of SPC24:SPC25. CENP-T was fluorescently labeled with FAM (wt) or TMR (mutants) and monitored specifically during chromatography and after SDS-PAGE. Red asterisks indicate mutated phosphorylation sites.

Thr85 or that was not phosphorylated, did not form a complex with SPC24:SPC25 (*Figure 2*, red and black traces).

Using a similar experimental setup, we determined that CENP-T also binds full-length NDC80Cs in a manner that depends on the phosphorylation of Thr11 and Thr85 (*Figure 3A* and *Figure 3—figure supplement 1*). We next set out to visualize these assemblies directly by electron microscopy (EM) after low-angle metal shadowing. This technique enhances the contrast of the coiled-coil regions of NDC80C, enabling a characterization of the overall dimensions of complexes containing NDC80C. The coiled-coil of NDC80C spans 51 nm and the NDC80:NUF2 - SPC24:SPC25 end-to-end length is 62 nm (corrected for an estimated 3 nm shadow contribution at both ends) (*Figure 3B* and *Figure 3—figure supplement 2*). This is slightly longer than the 57 nm of yeast NDC80C previously determined using the same technique (*Wei et al., 2005*). The difference might be explained by the use of full-length human NDC80C whereas the used yeast version lacked the N-terminal 100 residues of the Ndc80p subunit, although interspecies or technical differences cannot be excluded. The low-resolution and the flexible appearance of the NDC80C prevent unambiguous assignment of the SPC24:SPC25 and the NDC80:NUF2 modules of the complexes, although a clear kink in the coiled coil, previously attributed to a sequence insertion in NDC80 (*Ciferri et al., 2008*) marks NDC80:NUF2 in a number of cases (*Figure 3B* and *Figure 3—figure supplement 2*).

Metal-shadowed particles from an early eluting size-exclusion chromatography (SEC) fraction of the CENP-T:NDC80C mixture had a more heterogeneous appearance than NDC80C in isolation (*Figure 3B* and *Figure 3—figure supplements 3* and *4*). We were able to distinguish at least two different kinds of complexes, containing either one or two NDC80Cs. Though it is not possible to ascertain if complexes with one discernible NDC80 contain CENP-T, those containing two NDC80Cs are presumably assembled on phosphorylated CENP-T. This interpretation is further supported by the distance between the base of the complexes and the characteristic kinks in the coiled coil of NDC80:NUF2 (*Figure 3B* and *Figure 3—figure supplement 4*). The apparent orientation of the two CENP-T-bound NDC80Cs widely varies, presumably because the CENP-T$^{11-85}$ stretch spanning the two NDC80C binding sites is flexible. In brief, we visualized how CENP-T recruits two NDC80Cs. The SPC24:SPC25 modules of these complexes are coordinated by the relative proximity of the phosphorylated Thr11 and Thr85 of CENP-T, but the microtubule-binding CH domains can be well over 100 nm apart.

## CENP-T phosphorylated at Ser201 stoichiometrically binds the MIS12 complex

It had previously been shown that MIS12C and the N-terminal region of CENP-T bind SPC24:SPC25 in a competitive manner (*Schleiffer et al., 2012*; *Nishino et al., 2013*). Consistent with these previous reports, MIS12C bound SPC24:SPC25, displacing CENP-T$^{2-101}$, already at equimolar concentrations, indicative of higher binding affinity (unpublished data). In apparent contradiction with these observations, however, it has also been reported that the phosphorylation of human CENP-T on Thr195, within a weakly conserved CDK1 consensus motif (*Figure 4A*), contributes to the recruitment of MIS12C to CENP-T in vivo (*Rago et al., 2015*). Furthermore, the CENP-T$^{200-230}$ region has been identified as a crucial determinant of the CENP-T:MIS12C interaction in vivo (*Rago et al., 2015*), indicating that residues downstream of Thr195 are also important for MIS12C binding.

Thr195 is flanked by a well-conserved downstream motif responding to the consensus **PXSVXR**, where X are positions where significant sequence variability is observed (*Figure 4A*). This motif conforms to a recently identified non-consensus CDK1 substrate signature (*Suzuki et al., 2015b*). Furthermore, the presence of Pro at the −2 position relative to the phospho-acceptor site has been previously recognized as a favorable signature for CDK1 phosphorylation, while Val at the +1 position is among the least unfavorable residues (*Alexander et al., 2011*). Both Thr195 and Ser201 have been previously identified as mitotic CENP-T phosphorylation sites (*Kettenbach et al., 2011*), and we confirmed that these residues are also found phosphorylated in CENP-T pulled down with a specific antibody from mitotic HeLa cells (*Figure 4—figure supplement 1*). Both sites were effectively phosphorylated by CDK1 in vitro (*Table 1* and *Figure 4—figure supplement 1*).

To assess if CENP-T binds directly to MIS12C, and if phosphorylation of Thr195 or Ser201 is important for this interaction, we generated synthetic fluorescent peptides encompassing residues CENP-T$^{184-215}$ and containing phosphorylated Thr195 or Ser201, incubated them with MIS12C, and analyzed the mixture by SEC. The peptide carrying phosphorylated Ser201 bound directly to

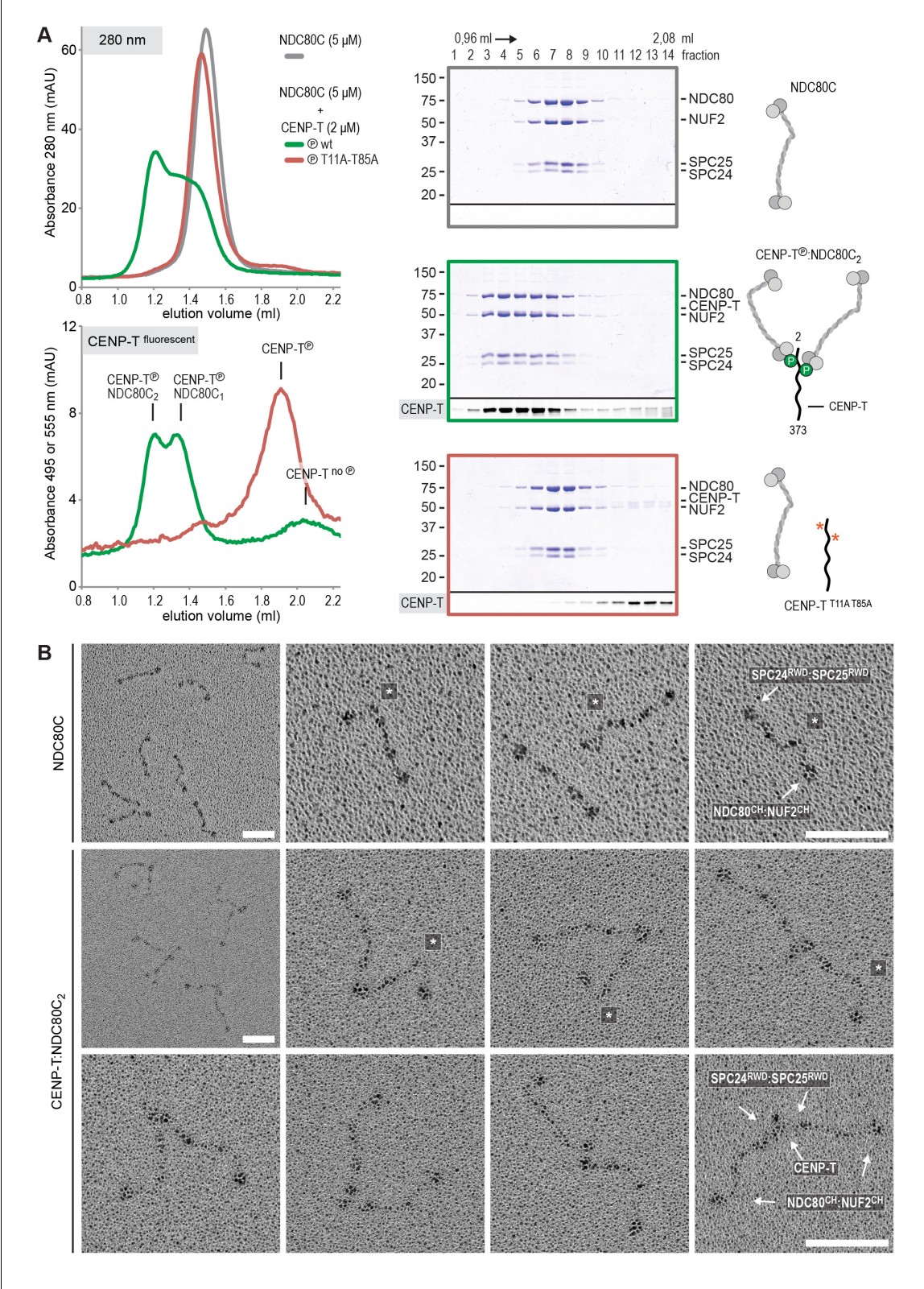

**Figure 3.** Phosphorylated CENP-T recruits two full-length NDC80 complexes. (**A**) Analytical size-exclusion chromatography (Superose 6 5/150) and SDS-PAGE show that CENP-T$^{2-373}$ phosphorylated by CDK1:Cyclin B at positions T11 and T85 can bind two full-length NDC80 complexes. The mixture of CENP-T and NDC80C contained NDC80C, CENP-T:NDC80C, as well as CENP-T:NDC80C$_2$ species (green trace). The specific monitoring of fluorescently labeled CENP-T was used to distinguish these species. Red asterisks indicate mutated phosphorylation sites. Analysis of the single T11A

*Figure 3 continued on next page*

*Figure 3 continued*

and T85A CENP-T mutants and additional controls are included in *Figure 3—figure supplement 1*. (B) NDC80C (top row) and CENP-T:NDC80C$_2$. (middle and bottom rows) were visualized by electron microscopy after glycerol spraying and low-angle platinum shadowing. Asterisks mark the kink in the NDC80:NUF2 coiled coil region. The first micrographs show a representative field of view at a lower magnification. All scale bars represent 50 nm. More micrographs of NDC80C and CENP-T:NDC80C$_2$ as well as sample preparation information are included in *Figure 3—figure supplements 2–4*.

The following figure supplements are available for figure 3:

**Figure supplement 1.** Phosphorylated CENP-T recruits two full-length NDC80 complexes.
**Figure supplement 2.** NDC80C gallery and measurements.
**Figure supplement 3.** Sample preparation for low-angle metal shadowing and EM.
**Figure supplement 4.** Gallery and measurements of CENP-T:NDC80C$_2$.

MIS12C in vitro, whereas the equivalent peptide containing phosphorylated Thr195 did not (*Figure 4B*). The importance of CENP-T phosphorylation at position Ser201 was recapitulated for the longer CENP-T$^{2-373}$ construct, where mutation of this residue was shown to abrogate binding to MIS12C (*Figure 4C* and *Figure 4—figure supplement 2*). Alanine mutation of the phospho-targets Thr11 and Thr85 (*Figure 4C*) or Thr195 (*Figure 4—figure supplement 3*) did not affect the binding between CENP-T and MIS12C. Phosphorylation of CENP-T at position Ser201 is thus both required and sufficient to mediate the MIS12C:CENP-T interaction in vitro. Although our results leave open the possibility that Ser201 is phosphorylated in vitro by a contaminating kinase, rather than by CDK1:Cyclin B, several factors contribute to mitigate this concern. First, the CDK1:Cyclin B sample we used is highly pure (*Figure 4—figure supplement 4A*) and devoid of other Ser/Thr kinases, as judged by mass spectrometry (unpublished data). Second, phosphorylation of CENP-T Ser201 was also observed using a commercial CDK1:Cyclin B preparation (*Figure 4—figure supplement 3*). Third, the addition of the CDK1 specific inhibitor RO-3306 abolished the overall phosphorylation of CENP-T by CDK1:Cyclin B (as judged by the migration of CENP-T on SDS-PAGE, *Figure 4—figure supplement 4B*). Finally, RO-3306 prevented the specific phosphorylation of CENP-T at Ser201 and inhibited the CENP-T:MIS12C interaction, *Figure 4—figure supplement 4C*).

We further characterized the interaction between CENP-T and MIS12C using sedimentation velocity analytical ultracentrifugation (SV-AUC) experiments. Both MIS12C and CENP-T sedimented with relatively high frictional ratios, indicative of an elongated shape or of lack of a defined conformation, respectively (*Figure 4D*). To analyze the sedimentation of the CENP-T:MIS12C mixture, we specifically monitored absorbance of the fluorescently labeled CENP-T and thus analyzed sedimentation of CENP-T:MIS12C without detecting MIS12C. The CENP-T:MIS12C complex sedimented with a frictional ratio of 2.4S and a determined mass (159 kDa), in close agreement with the theoretical mass of a 1:1 complex (162 kDa, unphosphorylated forms) (*Figure 4D*).

## Phosphorylation of CENP-T Ser201 is important for MIS12C recruitment in vivo

To analyze how CENP-T phosphorylation at various positions affects kinetochore recruitment of NDC80C and MIS12C in vivo, we adapted a previously established assay (*Gascoigne et al., 2011*) and tethered lac repressor (LacI)-EGFP-CENP-T$^{2-373}$ fusion proteins to an ectopically integrated lac operon (*lacO*) array in U2OS cells. As previously reported (*Gascoigne et al., 2011*; *Rago et al., 2015*), NDC80C and MIS12C were recruited to CENP-T loci in prometaphase and not in interphase (*Figure 5A*), in agreement with the observed CDK1-dependent recruitment of these complexes to CENP-T in vitro (*Figures 3* and *4*). The NDC80C binding mutant CENP-T$^{T11A-T85A}$ binds MIS12C in vitro (*Figure 4C*), but in the context of a LacI-EGFP fusion, it failed to recruit not only NDC80C, but also MIS12C, to the *lacO* locus, as previously reported (*Rago et al., 2015*) (*Figure 5B*). CENP-T$^{S201A}$, on the other hand, recruited NDC80C to the *lacO* array, but no longer interacted with MIS12C (*Figure 5B*). This result is consistent with the requirement for phosphorylation of Ser201 for MIS12C recruitment in vitro (*Figure 4*), and with the previous identification of the CENP-T$^{200-230}$

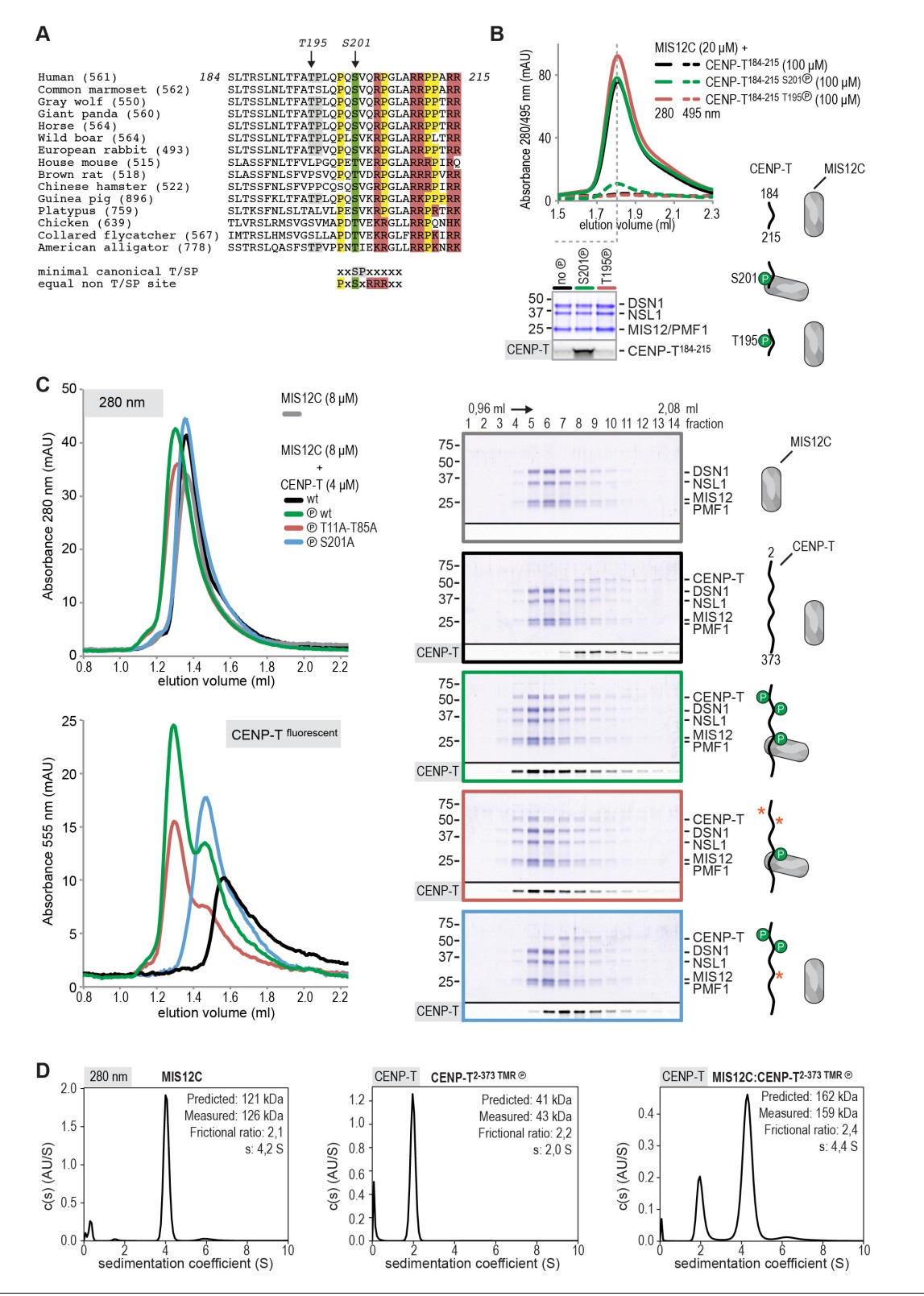

**Figure 4.** CENP-T phosphorylated by CDK1 at position S201 binds the MIS12 complex. (**A**) Multiple sequence alignment of the CENP-T[184-215] region. The CENP-T length in the various species, as well as the T195 and S201 phosphosites are indicated. The indicated non-canonical CDK1 recognition site, to which S201 and neighboring residues conform perfectly, and the shown CDK1 minimal canonical T/SP site were found to be equally good CDK1 substrates (**Suzuki et al., 2015b**). (**B**) Analytical size-exclusion chromatography (Superose 6 5/150 increase) shows that CENP-T[184-215] binds to MIS12C

*Figure 4 continued on next page*

*Figure 4 continued*

when phosphorylated at position S201. Dashed traces indicate absorption at 495 nm and represent peptide that co-elutes with MIS12C. Free peptides have a retention volume of 2.6 ml. Peak fractions were analyzed by Coomassie staining and in-gel fluorescence after SDS-PAGE. (C) Analytical size-exclusion chromatography (Superdex 200 5/150 increase) and SDS-PAGE show that CENP-T$^{2-373}$ phosphorylated by CDK1:Cyclin B at S201 binds to MIS12C. Red asterisks indicate mutated phosphorylation sites. Additional controls are included in *Figure 4—figure supplement 2*. (D) Sedimentation velocity analytical ultracentrifugation of MIS12C, CENP-T, and MIS12C:CENP-T show that MIS12C and CENP-T bind in a 1:1 complex. MIS12C was monitored by its absorbance at 280 nm whereas CENP-T and the CENP-T:MIS12C mixture were followed using the absorbance of the fluorescently labeled CENP-T$^{TMR}$ at 555 nm.

The following figure supplements are available for figure 4:

**Figure supplement 1.** CENP-T phosphorylation by CDK1:Cyclin B at positions T195 and S201.

**Figure supplement 2.** CENP-T phosphorylated by CDK1:Cyclin B at position S201 binds the MIS12 complex.

**Figure supplement 3.** CENP-T mutated at position T195 binds MIS12C.

**Figure supplement 4.** Purification of CDK1:Cyclin B and inhibition of CENP-T phosphorylation by RO-3306.

region as a crucial determinant of the CENP-T:MIS12C interaction in vivo (*Rago et al., 2015*). Interestingly, an alanine mutant of Thr195 of CENP-T prevented recruitment of both the MIS12C and the NDC80C to the ectopic *lacO* site, even if this residue is dispensable for MIS12C or NDC80C binding in vitro (*Figure 4B*, *Figure 4—figure supplement 3*, and *Figure 5B*). We therefore speculate that Thr195 contributes to the efficiency of phosphorylation or to the stability of CENP-T in vivo, an effect that is not recapitulated in vitro, where this residue appears to be dispensable for MIS12C binding. Overall, our data underline the importance of Ser201 of CENP-T as a MIS12C binding site both in vitro and in vivo.

## CENP-T recruits one MIS12 complex and three NDC80 complexes

To test directly if MIS12C that is bound to phosphorylated CENP-T retains its ability to bind NDC80C, we returned to our reconstituted system and incubated different constructs of CENP-T with MIS12C and a molar excess of NDC80C. Phosphorylated CENP-T integrated in large complexes whose elution volume from the SEC column was smaller than that of MIS12:NDC80 complexes. Our set of CENP-T mutants enabled a systematic analysis of these CENP-T:MIS12C:NDC80C assemblies. In addition to recapitulating observations made for CENP-T:NDC80C (*Figure 3*) and CENP-T: MIS12C (*Figure 4*), these experiments showed that CENP-T with mutated NDC80C-binding motifs (T11A and T85A) still associated with MIS12C:NDC80C (*Figure 6A*, red trace). This association strictly depended on the phosphorylation of CENP-T at Ser201 because the CENP-T triple mutant (T11A, T85A, S201A) no longer formed a complex with MIS12C or NDC80C (*Figure 6A*, purple trace). Collectively, our results demonstrate that 1) CENP-T$^{wt}$ associates with MIS12C and NDC80C in 1:1:3 ratio; 2) CENP-T$^{T11A}$ or CENP-T$^{T85A}$ associate with MIS12C and NDC80C in 1:1:2 ratio; and 3) CENP-T$^{T11A-T85A}$ associates with MIS12C and NDC80C in 1:1:1 ratio. The behavior of the complete set of CENP-T mutants upon incubation with MIS12C and/or NDC80C is displayed in *Figure 6—figure supplement 1*). Thus, MIS12C that interacts with CENP-T phosphorylated at Ser201 can also interact with NDC80C. This complex was also able to incorporate KNL1C, indicating that it can bind an entire KMN assembly, as is the case for MIS12C bound to CENP-C (*Petrovic et al., 2016*) (*Figure 6—figure supplement 2*). If KNL1C is recruited to the outer kinetochore in a CENP-T-mediated and phosphorylation-regulated manner in vivo, and if this contributes to spindle assembly checkpoint signaling, remains to be tested.

## Visualization of the outer kinetochore proteins assembled on CENP-T

To analyze the overall appearance of CENP-T:MIS12C:NDC80C complexes, we incubated isolated CENP-T:MIS12C complexes with a molar excess of NDC80C, separated the resulting mixture by SEC (*Figure 3—figure supplement 3*), and inspected both a high-molecular weight fraction and a later-eluting fraction by EM after low-angle metal shadowing. As previously described for complexes that

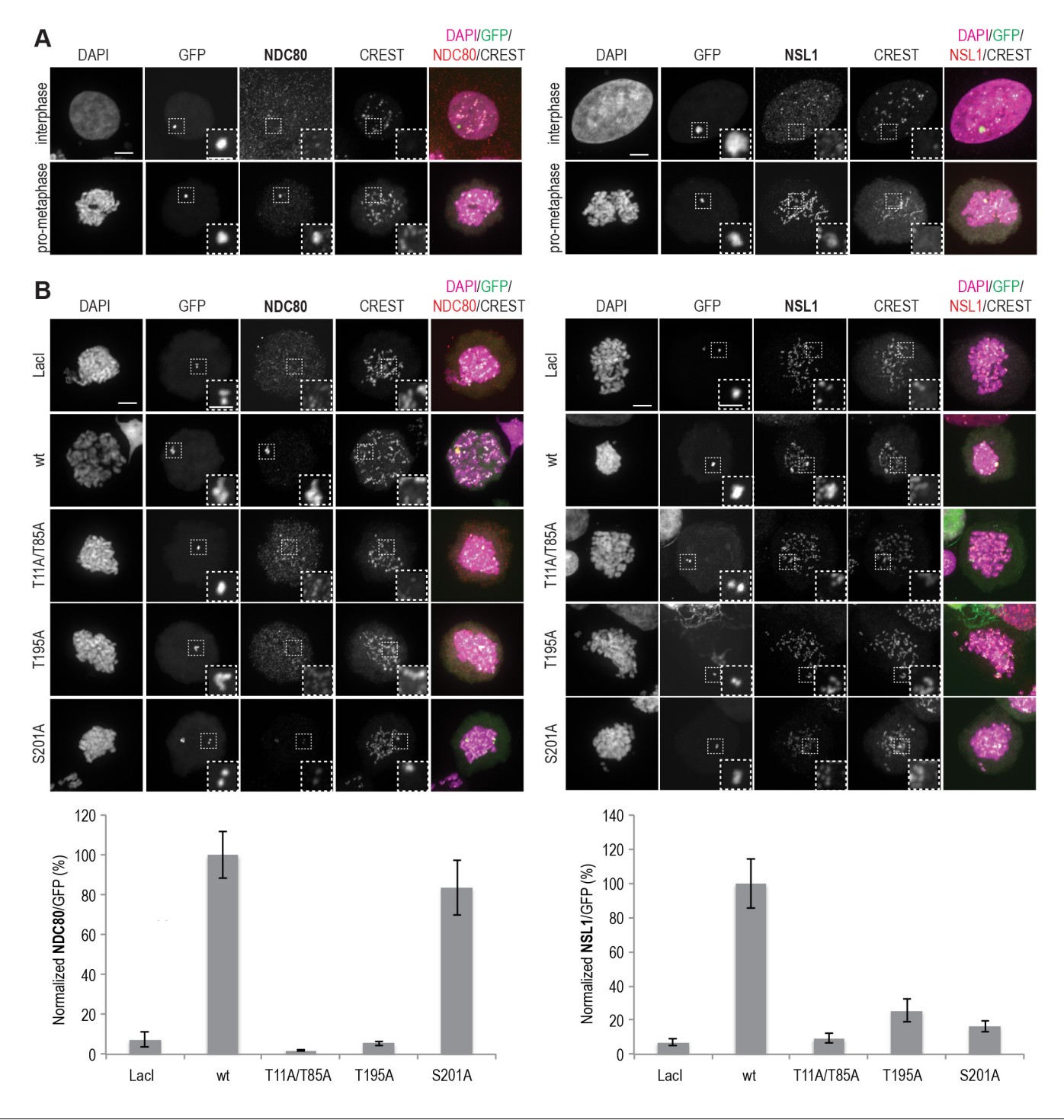

**Figure 5.** MIS12C and NDC80C are recruited to ectopically localized CENP-T in U2OS cells. (**A**) U2OS cells that contain a lacO array recruit transiently expressed LacI-EGFP-CENP-T$^{2-373}$ fusion constructs throughout the cell cycle. NDC80C (antibody against NDC80) and MIS12C (antibody against NSL1) are recruited to ectopic CENP-T loci in prometaphase, but not in interphase. Endogenous kinetochores are marked by CREST, DAPI was used to stain the DNA, and the EGFP signal indicates LacI-EGFP or LacI-EGFP-CENP-T constructs and marks the position of the lacO array. Scale bars represent 5 μm for the overview images and 2 μm for the magnified insets. (**B**) Representative prometaphase cells expressing LacI-EGFP or LacI-EGFP-CENP-T (wild-type or mutant) fusion constructs. (**C**) The mean NDC80C/EGFP and MIS12C/EGFP signal intensity ratios were determined from three independent experiments and normalized to the ratio for the wild-type construct. A total of 12–27 cells were analyzed per LacI-EGFP-CENP-T fusion construct and error bars indicate standard error of mean.

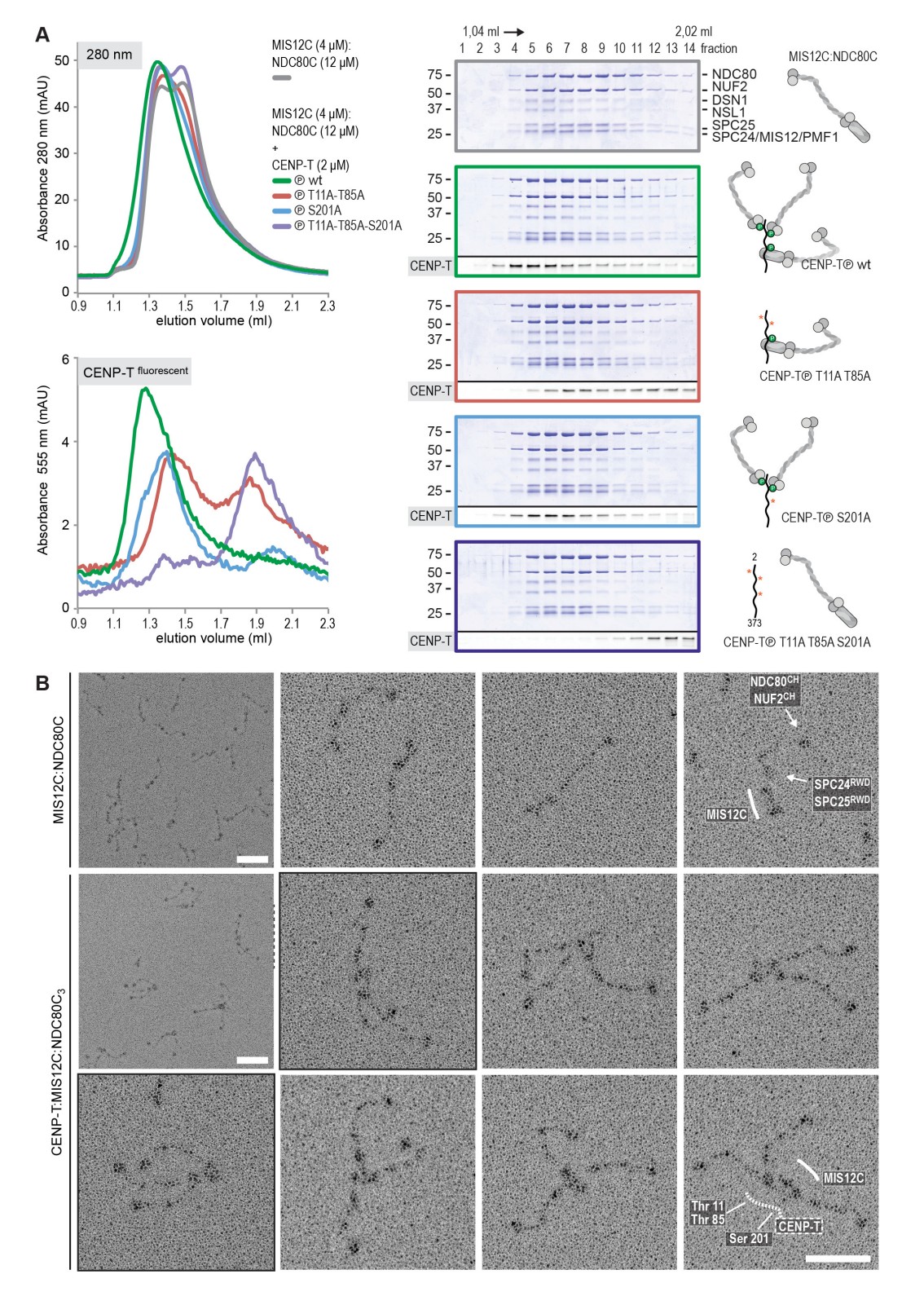

**Figure 6.** Reconstitution and visualization of CENP-T:MIS12C bound to three full-length NDC80 complexes. (**A**) Analytical size-exclusion chromatography (Superose 6 5/150 increase) and SDS-PAGE show that CENP-T$^{2-373}$ phosphorylated by CDK1:Cyclin B can bind MIS12C and three full-length NDC80 complexes. The triple mutation of phospho-sites T11, T85, and S201 is sufficient to prevent CENP-T from interacting with NDC80C and MIS12C:NDC80C (purple). Red asterisks indicate mutated phosphorylation sites. Fluorescently labeled CENP-T was monitored specifically during

*Figure 6 continued on next page*

*Figure 6 continued*

chromatography and after SDS-PAGE. The analysis of the complete set of CENP-T mutants is included as *Figure 6—figure supplement 1.* (B) MIS12C: NDC80C (top row) and CENP-T:MIS12C:NDC80C$_3$ (middle and bottom rows) were visualized by electron microscopy after glycerol spraying and low-angle platinum shadowing. MIS12C forms a rod-like extension in the MIS12C:NDC80C complexes and can in some cases also be distinguished as a module in CENP-T:MIS12C:NDC80C$_3$ assemblies. In those cases, CENP-T T11 and T85 are positioned at the base of the other two NDC80Cs (as annotated in the bottom right micrograph). The first micrographs show a representative field of view at a lower magnification. Scale bars represent 50 nm. Sample preparation is described in *Figure 3—figure supplement 3*. More micrographs of MIS12C:NDC80C and CENP-T:MIS12C:NDC80C$_3$ are included in *Figure 6—figure supplements 3–4*.

The following figure supplements are available for figure 6:

**Figure supplement 1.** Supercomplex formation depends on the phosphorylation of CENP-T T11, T85, and S201.

**Figure supplement 2.** CENP-T phosphorylated at position S201 can recruit the KMNZ network.

**Figure supplement 3.** Gallery and measurements of MIS12C:NDC80C.

**Figure supplement 4.** Galleries of CENP-T:MIS12C:NDC80C.

were analyzed by electron microscopy after negative staining (*Petrovic et al., 2010*; *Screpanti et al., 2011*; *Petrovic et al., 2014*), analysis of metal-shadowed MIS12C:NDC80C showed that MIS12C forms a rod-like 18 nm extension at the SPC24:SPC25 side of NDC80C (*Figure 6B* and *Figure 6—figure supplement 3*). In agreement with our biochemical analysis, direct visualization of CENP-T:MIS12C:NDC80C assemblies highlighted that CENP-T can coordinate MIS12C and a total of three NDC80Cs into a single complex (*Figure 6B* and *Figure 6—figure supplement 4*). Such assemblies were rare in the later eluting fraction of the same SEC run. In several cases, the MIS12C:NDC80C module could be distinguished from the two NDC80Cs that are recruited to CENP-T phosphorylated at positions Thr11 and Thr85, which resembled the isolated CENP-T:NDC80C$_2$ complexes (*Figure 6B*,e.g. bottom right).

In a small number of cases, the core of MIS12C:CENP-T seemed to be more complex, resulting in assemblies with more density in the middle and four, five, or six NDC80Cs at the periphery (*Figure 6—figure supplement 4*). Whether this oligomerization represents a biologically relevant multimerization potential of the outer kinetochore remains to be tested. We also observed CENP-T: MIS12C:NDC80$_2$ assemblies in which one NDC80C is directly recruited to phosphorylated CENP-T via Thr11 or Thr85 and one NDC80C binds CENP-T:MIS12C (*Figure 6—figure supplement 4*).

## CENP-T and CENP-C are competitive binders of the MIS12 complex

The observation that CENP-T and MIS12C interact directly raises the possibility that the CENP-C- and CENP-T-mediated outer kinetochore recruitment pathways are integrated into a single CDK1: Cyclin B regulated pathway. Inspired by our recent structure-functional analysis of the human CENP-C:MIS12C complex (*Petrovic et al., 2016*), we set out to test this hypothesis further. CENP-C binds at the interface of Head1 and the helical connector, two structural elements of MIS12C. This interface is stabilized by Tyr8, Phe12, and Phe13 in the α0 helix of the Mis12 subunit and mutation of these three aromatic residues to alanine (MIS12C$^{3xA}$) abolished the binding of MIS12C with CENP-C (*Petrovic et al., 2016*). MIS12C$^{3xA}$ was also unable to interact with CDK phosphorylated CENP-T (*Figure 7A*), suggesting that CENP-T and CENP-C bind to similar sites of MIS12C.

This was confirmed in fluorescence polarization experiments, in which both CENP-C$^{1-71}$ and Ser201 phosphorylated CENP-T$^{195-215}$ displaced fluorescently labeled CENP-C$^{1-21}$ from MIS12C (*Figure 7B*). The relatively high concentration of phosphorylated CENP-T$^{195-215}$ peptide required to achieve this displacement reflect the tight MIS12C:CENP-C$^{1-21}$ binding ($K_d$ 128 nM) and the lower MIS12C:CENP-T$^{192-215}$ binding affinity ($K_d$ ~30 μM) (*Figure 7—figure supplement 1*). Longer CENP-C$^{2-545}$ and CENP-T$^{2-373}$ fragments also bound MIS12C in a competitive manner (*Figure 7C*). As expected, this strictly depended on the ability of CENP-C to bind MIS12C since the CENP-C$^{K10A-Y13A}$ MIS12C binding mutant (*Screpanti et al., 2011*) did not disrupt the preformed CENP-T:MIS12C complex (*Figure 7C*). These results indicate that the binding sites of CENP-C and CENP-T on

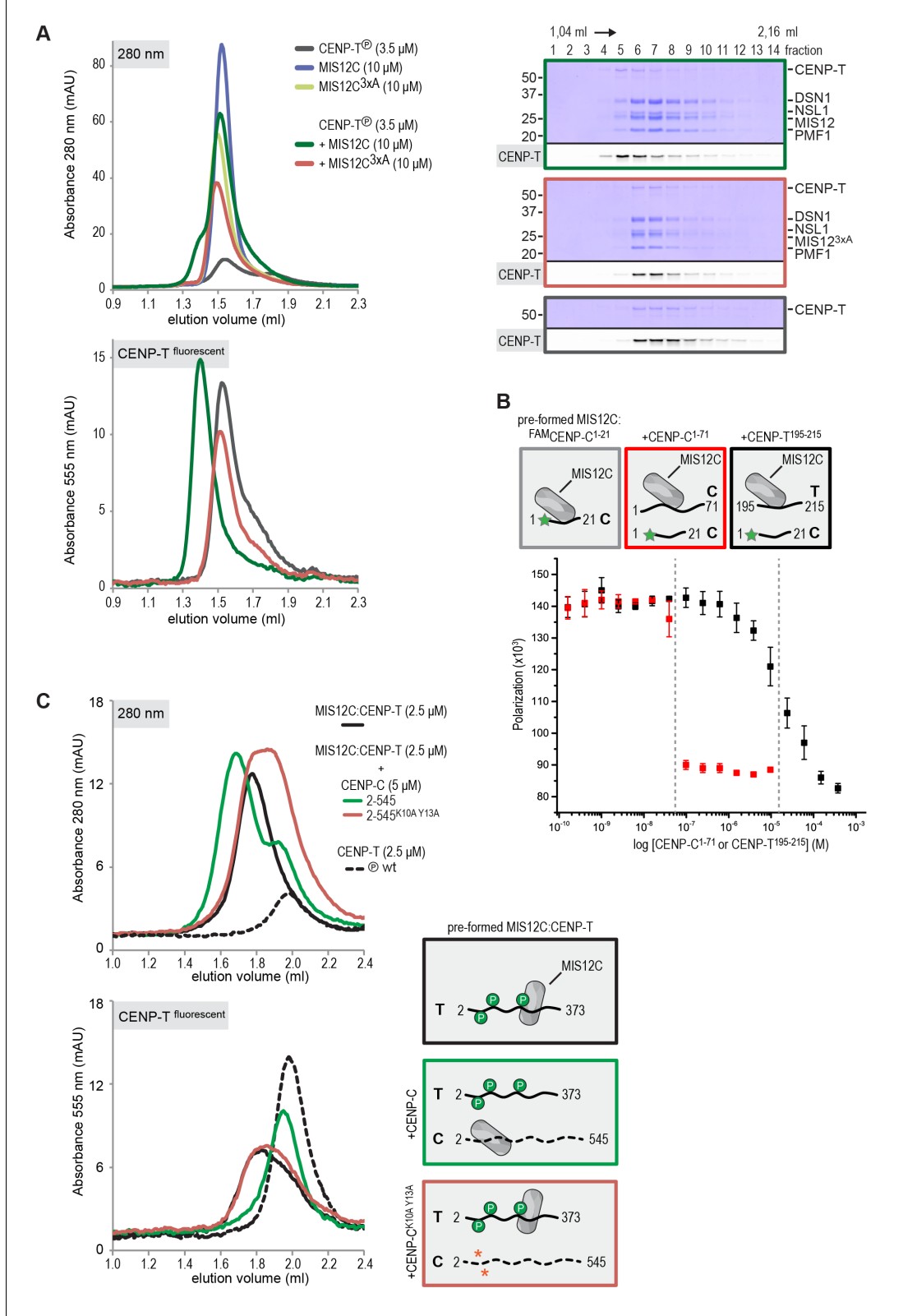

**Figure 7.** CENP-T and CENP-C are competitive binders of the MIS12 complex. (**A**) The binding of phosphorylated CENP-T to MIS12C[Nano], a version of MIS12C designed for structural analysis (*Petrovic et al., 2016*), is abolished by the mutation of Tyr8, Phe12, and Phe13 to alanine in the MIS12 subunit. (**B**) Fluorescent polarization was used to determine the displacement of CENP-C[1-21] from MIS12C[Nano] upon the addition of either CENP-C[1-71] or, at higher concentrations, CENP-T[195-215]. (**C**) Analytical size-exclusion chromatography (Superdex 200 5/150) shows that phosphorylated CENP-T is

*Figure 7 continued on next page*

*Figure 7 continued*

displaced from MIS12C when incubated with CENP-C, but not when incubated with a version of CENP-C that cannot bind to MIS12C. Fluorescently labeled CENP-T was monitored specifically during chromatography. Red asterisks indicate K10A and Y13A mutations in CENP-C. See *Figure 7—figure supplement 2* for the binding of CENP-C and CENP-C[K10A Y13A] to MIS12C.

The following figure supplements are available for figure 7:

**Figure supplement 1.** Phosphorylated CENP-T[192-215] binds to MIS12C.

**Figure supplement 2.** CENP-C with K10A and Y13A mutations does not bind to MIS12C.

MIS12C are at least partly overlapping, thus predicting that two distinct MIS12Cs are recruited to CENP-C and CENP-T.

## Discussion

Full assembly of the outer kinetochore is restricted to mitosis and is tightly regulated, but the molecular requirements for this regulation remain poorly understood. Here, we have reported a thorough dissection of the role of CENP-T and its phosphorylation by CDK1 in this process. Depletion of CENP-T almost completely abolishes the ultrastructure of the outer kinetochore (*Hori et al., 2008*) and phenocopies the effects of NDC80C depletion (*DeLuca et al., 2005*), underscoring the importance of CENP-T mediated recruitment of NDC80C to mitotic kinetochores. This role is further illustrated by the observation that CENP-T depletion lowers NDC80C levels at mitotic kinetochores more than CENP-C depletion (*Gascoigne et al., 2011*; *Suzuki et al., 2015a*), and by experiments showing that replacement of the outer kinetochore binding domain of CENP-C with that of CENP-T supports chromosome segregation in cells depleted of CENP-C (*Suzuki et al., 2015a*).

We reconstituted the assembly of outer kinetochore modules using purified components and determined how CDK-dependent phosphorylation of CENP-T controls the stoichiometry of NDC80C. The analysis of our synthetic outer kinetochores by SEC, SV-AUC, and EM revealed that one copy of CENP-T recruits two NDC80Cs via phosphorylated CENP-T[T11] and CENP-T[T85] and an entire KNL1C:MIS12C:NDC80C via phosphorylated CENP-T[S201]. Because the binding of CENP-C and CENP-T to MIS12C is competitive, these findings support a model in which CENP-C and CENP-T act in parallel to recruit two MIS12Cs, two KNL1Cs, and up to four NDC80Cs in a CDK-regulated manner (*Figure 8*).

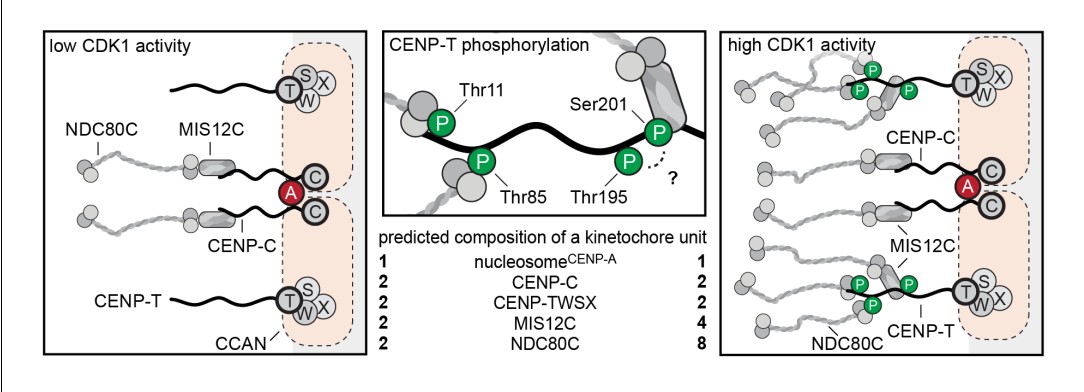

**Figure 8.** Molecular basis of stoichiometric assembly of microtubule binders in the outer kinetochore. Schematic representation of the results presented here and their implications. CDK activity results in the recruitment of NDC80C and MIS12C:NDC80C to CENP-T. The predicted stoichiometry of a single kinetochore unit is indicated. In the middle panel, we summarize the role of T11, T85, and S201 in recruitment of NDC80C and MIS12C, and suggest that T195 may contribute to phosphorylation of S201 in vivo.

Our finding that phosphorylated CENP-T can form a 1:1 complex with MIS12C is consistent with a series of in vivo studies that showed that CENP-T does not only bring NDC80C to the outer kinetochore, but is also involved in the recruitment of MIS12 (*Gascoigne et al., 2011*; *Gascoigne and Cheeseman, 2013*; *Hori et al., 2013*; *Kim and Yu, 2015*; *Rago et al., 2015*; *Suzuki et al., 2015a; Wood et al., 2016*). The molecular mechanism behind the contribution of CENP-T to MIS12C localization at the kinetochore, however, had remained unclear, and it appeared puzzling in view of observations that MIS12C and CENP-T bind to NDC80C in a competitive manner (*Schleiffer et al., 2012*; *Nishino et al., 2013*). A role for the CENP-T$^{200-230}$ region in MIS12C recruitment was shown in experiments of ectopic chromatin targeting of CENP-T fragments, which revealed that MIS12C was recruited to CENP-T$^{1-230}$ but not to CENP-T$^{1-200}$ (*Rago et al., 2015*). Our observation that the phosphorylation of Ser201 is essential for the CENP-T:MIS12C interaction in vitro is consistent with these previous observations (*Figure 4*). Phosphorylation of the weakly conserved CDK consensus site Thr195 is important for the CENP-T:MIS12C interaction in vivo (*Figure 5B* and *Rago et al., 2015*), but is not required in vitro (*Figure 4—figure supplement 3*). A possible explanation for these results is that the phosphorylation of kinetochore-associated CENP-T at Thr195 is required for the phosphorylation of Ser201 and the subsequent binding of MIS12C. Multisite phosphorylation networks are important for the regulation of mitosis by CDK1 kinase (*Kõivomägi et al., 2011*) and future experiment will be required to understand the spatial and temporal regulation of CENP-T phosphorylation in vivo. Why the CENP-T T195A mutation also interfered with CENP-T:NDC80C recruitment to ectopic *lacO* loci in our hands (*Figure 5B*) but not in a previous study (*Rago et al., 2015*) is unclear, but is consistent with a role of this residue in the overall phosphorylation or stability of CENP-T. Our observation that the double mutant CENP-T$^{T11A-T85A}$ was unable to recruit MIS12C to *lacO* loci agrees with previous observations (*Rago et al., 2015*), and indicates that binding of NDC80C to CENP-T facilitates further recruitment of MIS12C (*Figure 5B*). The molecular basis for this phenomenon is currently unclear and will require further investigation. So far, we have been unable to recapitulate this effect with purified components.

Our reconstitution experiments provide a molecular explanation for the observed CDK-dependent recruitment of NDC80C to kinetochores (*Gascoigne et al., 2011*) and show that one CENP-T molecule can recruit one MIS12C and up to three NDC80Cs. Because the pseudo-symmetry of the CENP-A nucleosomes allows its binding to two CCAN modules (*Weir et al., 2016*), we propose that mitotic phosphorylation events trigger the recruitment of four MIS12Cs (and KNL1C) and up to eight NDC80Cs per CENP-A mononucleosome (*Figure 8*). This extrapolation of our reconstitution experiments is in line with a series of quantitative analyses of kinetochore subcomplexes by immunofluorescence microscopy. Copy numbers of CENP-C, CENP-T, MIS12C, and NDC80C were most accurately determined for metaphase kinetochores in a study that combined RNA interference, ectopic and endogenous kinetochore recruitment, and an elegant CENP-C/T chimera protein design (*Suzuki et al., 2015a*). This analysis revealed that, on average, 215 copies of CENP-C and 72 copies of CENP-T recruit a total of 244 NDC80Cs and 151 MIS12Cs to a kinetochore. However, only 82 (of the 215) CENP-C proteins engage in binding MIS12C, whereas the remaining CENP-C does not contribute to outer kinetochore recruitment. CENP-T is responsible for the recruitment of the remaining MIS12Cs and NDC80Cs (*Suzuki et al., 2015a*). These results correlate well to our description of 1:1:1 CENP-C:MIS12C:NDC80C and 1:1:2 or 1:1:3 CENP-T:MIS12C:NDC80C assemblies.

As stated in the introduction, NDC80C multivalency may have profound implications for microtubule binding (*Hill, 1985*; *Zaytsev et al., 2014*). Our study paves the way to the reconstitution of synthetic kinetochores with controllable numbers of NDC80 complexes, which will in turn allow us to probe the importance of multivalency for force production by kinetochores. Whether the CDK-regulated localization of multiple NDC80Cs at the outer kinetochore has an effect on the function of kinetochores to signal the microtubule-attachment state through the spindle checkpoint control also remains to be addressed. In this perspective, it is interesting to note that NDC80C recruited by CENP-C:MIS12C or by CENP-T have been proposed to have different effects on the checkpoint response (*Kim and Yu, 2015*; *Samejima et al., 2015*).

The visualization of the reconstituted 1:1:3 CENP-T:MIS12C:NDC80C complexes by electron microscopy after low-angle metal shadowing highlighted how outer kinetochores assembled on phosphorylated CENP-T can span distances of well over 100 nm. It also showed the varying orientation of the coordinated NDC80Cs relative to each other, presumably caused by the flexibility of the CENP-T regions between the NDC80C and MIS12C binding sites. The flexibility of CENP-T as well

as the stretching of CENP-T upon kinetochore-microtubule contacts has been shown before: the N- and C-termini of tagged CENP-T were ~30 nm apart in the presence of tension (MG132 treated cells) but only ~4 nm apart in the absence of tension (nocodazole treated cells), as determined by immunofluorescence and immuno-electron microscopy (*Suzuki et al., 2011*). Moreover, the N-terminal tail of CENP-T was seen as a 25 ± 13 nm long flexible extension of the CENP-TW histone-fold domain by high speed atomic force microscopy (*Suzuki et al., 2011*).

The likely interpretation of these results is that the intrinsically flexible N-terminal region of CENP-T stretches when the multiple NDC80Cs it coordinates bind microtubules, while its C-terminal HFD remains anchored at the inner kinetochore. In the future, it will be important to test how the observed CENP-T:MIS12C:KNL1:NDC80C$_3$ complexes structurally rearrange in the presence of microtubules and if the presence of multiple NDC80Cs contributes to the stabilization of the kinetochore-microtubule interface observed in the presence of tension (*Nicklas and Koch, 1969*; *Akiyoshi et al., 2010*). Low affinity, cooperative interactions might acquire the highest relevance under these conditions, and biochemical reconstitution is a powerful means to identify them and address their possible importance.

## Materials and methods

### Protein expression and purification

cDNAs encoding GST-CENP-T$^{2-101}$ and GST-CENP-T$^{2-373}$ with a C-terminal -LPETGG extension were subcloned into pGEX-6P-2rbs, a dicistronic derivative of pGEX6P, generated in house. Point mutations were introduced by PCR and all plasmids were verified by DNA sequencing. *E. coli* BL21(DE3)-Codon-plus-RIL cells containing the CENP-T encoding pGEX-6P-2rbs vector were grown at 37°C in Terrific Broth in the presence of Chloramphenicol and Ampicillin to an OD$_{600}$ of ~0.8. Protein expression was induced by the addition of 0.35 mM IPTG and cells were incubated ~14 hr at 20°C. All following steps were performed on ice or at 4°C. Cell pellets were resuspended in buffer A (20 mM Tris-HCl, pH 8.0, 300 mM NaCl, 10% (v/v) glycerol and 1 mM TCEP) supplemented with 0.5 mM PMSF, protease-inhibitor mix HP Plus (Serva) and DNaseI (Roche), lysed by sonication and cleared by centrifugation at 108,000*g* for 30 min. The cleared lysate was applied to a 5 ml GSTrap FF column (GE Healthcare) equilibrated in buffer A. The column was washed with approximately 50 column volumes of buffer A and bound protein was eluted in buffer A by cleavage of the GST-tag with PreScission protease for ~14 hr. The eluate was diluted 10-fold with buffer B (20 mM Tris-HCl pH 6.8, 1 mM TCEP) and applied to a 5 ml HiTrap Heparin HP column (GE Healthcare) pre-equilibrated in buffer B with 25 mM NaCl. Bound protein was eluted in buffer B using a linear gradient from 25 mM to 800 mM NaCl in 16 column volumes. Selected fractions were concentrated (Amicon; 10 kDa molecular weight cut-off) and either used for Sortase-mediated labeling (see below) or directly applied to an equilibrated Superdex 200 10/300 column (GE Healthcare). SEC was performed with a 20 mM Tris-HCl, pH 8.0, 150 mM NaCl, and 1 mM TCEP mobile phase at a flow rate of 0.4 ml/min, and the relevant fractions were pooled, concentrated, flash-frozen in liquid nitrogen, and stored at −80°C.

The four full-length components of the NDC80 complex (NDC80C; SPC25$^{6xHis}$) were combined on a pFL (*Fitzgerald et al., 2006*) or a pBIG1 (*Weissmann et al., 2016*) vector. Baculoviruses were generated in Sf9 insect cells and used for protein expression in Tnao38 insect cells (*Hashimoto et al., 2010*). Between 60 and 72 hr post-infection, cells were pelleted, washed in PBS, pelleted, and stored at −80°C until use. Cells were thawed and resuspended in buffer A (50 mM Hepes, pH 8.0, 200 mM NaCl, 5% v/v glycerol, 1 mM TCEP) supplemented with 20 mM imidazole, 0.5 mM PMSF, and protease-inhibitor mix HP Plus (Serva), lysed by sonication and cleared by centrifugation at 108,000*g* for 30 min. The cleared lysate was filtered (0.8 μM) and applied to a 5 ml HisTrap FF (GE Healthcare) equilibrated in buffer A with 20 mM imidazole. The column was washed with approximately 50 column volumes of buffer A with 20 mM imidazole and bound proteins were eluted in buffer A with 300 mM imidazole. Relevant fractions were pooled, diluted 5-fold with buffer A with 25 mM NaCl and applied to a 6 ml ResourceQ column (GE Healthcare) equilibrated in the same buffer. Elution of bound protein was achieved by a linear gradient from 25 mM to 400 mM NaCl in 30 column volumes. Relevant fractions were concentrated in 30 kDa molecular mass cut-off Amicon concentrators (Millipore) in the presence of an additional 200 mM NaCl and applied to a

Superdex 200 10/300 or a Superose 6 10/300 column (GE Healthcare) equilibrated in size-exclusion chromatography buffer (50 mM Hepes, pH 8.0, 250 mM NaCl, 5% v/v glycerol, 1 mM TCEP). Size-exclusion chromatography was performed under isocratic conditions at the recommended flow rates and the relevant fraction were pooled, concentrated, flash-frozen in liquid nitrogen, and stored at −80°C. Expression and purification of the full-length MIS12 complex (MIS12C; Dsn1[6His]) was identical but size-exclusion chromatography was performed in 50 mM Hepes, pH 8.0, 200 mM NaCl, 1 mM and no additional NaCl was added before the concentration of the relevant fractions of the ion-exchange chromatography step. MIS12C[Nano], designed for structural analysis (*Petrovic et al., 2016*), was used for the experiments shown in *Figure 7A*, *Figure 7B*, and *Figure 7—figure supplement 1*.

SPC24:SPC25 (*Ciferri et al., 2008*), CENP-C[1-71] (*Screpanti et al., 2011*) and CENP-C[2-545] (*Klare et al., 2015*) were expressed and purified as described.

Codon-optimized CDK1 and Cyclin B1 constructs were obtained from GeneArt (Life Technologies, Carlsbad, CA) and combined in a pBIG1A vector (*Weissmann et al., 2016*) with N-terminal tags on CDK1 (GST) and Cyclin B1 (hexahistidine). GST-CDK1 and 6His-Cyclin B1 were co-expressed in Tnao38 insect cells as described above and purified using glutathione sepharose (GE Healthcare) followed by size-exclusion chromatography using a HiLoad 16/60 Superdex 200 pg column (GE Healthcare) equilibrated in buffer containing 20 mM HEPES pH 7.5, 200 mM NaCl, 1 mM TCEP, and 5% glycerol.

## Sortase labeling

Purified *S. aureus* Sortase (*Guimaraes et al., 2013*) or the Sortase 5M mutant (*Hirakawa et al., 2015*) were used to label CENP-T[2-373]-LPETGG with GGGGK peptides with a C-terminally conjugated tetramethylrhodamine (TMR) or fluorescein amidite (FAM) (Genscript). Labeling was performed for ~14 hr at 4°C in the presence of 10 mM $CaCl_2$ using molar ratios of Sortase, CENP-T, and peptide of approximately 1:20:200. CENP-T was separated from Sortase and the unreacted peptides by size-exclusion chromatography as described above. Based on the absorbance at 280 nm and 495 nm (FAM) or 555 nm (TMR), a labeling efficiency of >90% was achieved.

## In vitro phosphorylation of CENP-T

CENP-T[2-101] fragments and CENP-T[2-373] used in *Figure 4—figure supplement 3* were in vitro phosphorylated by CDK1:Cyclin B (Millipore). In-house generated CDK1:CyclinB was used for all other in vitro phosphorylation experiments. Phosphorylation reactions were set up in size-exclusion chromatography buffer (20 mM Tris-HCl pH 8, 150 mM NaCl, and 1 mM TCEP) containing 100 nM CDK1: Cyclin B, 10 µM CENP-T substrate, 2 mM ATP, and 10 mM $MgCl_2$. Reaction mixtures were kept at 30°C for 30–60 min and then used in binding experiments.

## Identification of CENP-T phosphorylation by mass spectrometry

Liquid chromatography coupled with mass spectrometry was used to assess the phosphorylation status of in vitro and in vivo phosphorylated CENP-T. Flp-In T-REx HeLa cells carrying the EGFP transgene (*Krenn et al., 2012*) were maintained in DMEM (PAN Biotech) supplemented with 10% tetracycline-free FBS (Clontech) in the absence of doxycycline, penicillin and streptomycin (GIBCO) and 2 mM L-glutamine (PAN Biotech). Cells were treated with 330 nM nocodazole (Sigma) for 16 hr and mitotic cells were harvested by shake off and lysed by incubation in lysis buffer [150 mM KCl, 75 mM Hepes, pH 7.5, 1 mM EGTA, 1.5 mM $MgCl_2$, 10% glycerol, 90 U/ml benzonase (Sigma) and 0.1% NP-40 supplemented with protease inhibitor cocktail (Serva) and PhosSTOP phosphatase inhibitors (Roche)]. Clarified extracts were pre-cleared with protein A–Sepharose (CL-4B; GE Healthcare) for 1 hr at 4°C before a 2 hr incubation with anti-CENP-T/W antibody (*Basilico et al., 2014*) cross-linked to protein A-beads. Beads were washed twice with lysis buffer containing NP-40 0.2%, twice with lysis buffer without detergents and twice with water. Samples were reduced, alkylated and digested with LysC and Trypsin and prepared for mass spectrometry as previously described (*Rappsilber et al., 2007*). Immunoprecipitated samples were digested directly from the beads whereas in vitro phosphorylated CENP-T samples were digested in solution. Obtained peptides were separated on an EASY-nLC 1000 HPLC system (Thermo Fisher Scientific, Odense, Denmark) using a 45 min gradient from 5–60% acetonitrile with 0.1% formic acid and directly sprayed via a

nano-electrospray source in a quadrupole Orbitrap mass spectrometer (Q Exactive, Thermo Fisher Scientific) (*Michalski et al., 2011*). The Q Exactive was operated in a data-dependent mode acquiring one survey scan and subsequently ten MS/MS scans. To identify phospho-sites, the resulting raw files were processed with MaxQuant (version 1.5.2.18) searching for CENP-T$^{2-373}$ (in vitro samples) or the uniprot human database (in vivo samples) with deamidation (NQ), oxidation (M) and phosphorylation (STY) as variable modifications and carbamidomethylation (C) as fixed modification. A false discovery rate cut off of 1% was applied at the peptide and protein levels and as well on the site decoy fraction (*Cox and Mann, 2008*).

## Analytical size-exclusion chromatography

Proteins were mixed at the indicated concentrations, incubated on ice for at least one hour, spun for 15 min at 13,000 rpm at 4°C, and then analyzed by size-exclusion chromatography at 4°C using a ÄKTAmicro system (GE Healthcare) mounted with a column (Superdex 200 5/150, Superdex 200 5/150 increase, Superose 6 5/150, or Superose 6 5/150) equilibrated in size-exclusion chromatography buffer (20 mM Tris-HCl pH 8, 150 mM NaCl, and 1 mM TCEP) and operated at or near the recommended flow rate. Fractions of 50–100 µL were collected in 96-well plates and analyzed by SDS-PAGE. The absorbance of fluorescently labeled proteins was followed during chromatography using the ÄKTAmicro Monitor UV-900 (GE Healthcare) and after SDS-PAGE using a ChemiDoc MP system (Bio-Rad).

## Isothermal titration calorimetry

All samples were exchanged into fresh buffer (20 mM Tris-HCl, 150 mM NaCl and 1 mM TCEP). ITC measurements were performed at 20°C on an ITC200 microcalorimeter (GE Healthcare). In each titration, the protein in the cell (at a 10–30 µM concentration) was titrated with 25 × 4 µl injections (at 180 s intervals) of protein ligand (at 10-fold higher molar concentration). The following synthetic peptides (purity >95%, Genscript) were used Thr11 DST(p)PRTLLRRVLDTAYA, Thr85 EQT(p)PRTLLKNILLTAYA, Thr27 PRT(p)PRRPRSARAGARYA, and Thr47 TAS(p)PRKLSGQTRTIARYA. Injections were continued beyond saturation to allow for determination of heats of ligand dilution. Data were fitted by least-square procedure to a single-site binding model using ORIGIN software package (MicroCal).

## Fluorescence polarization

Fluorescence polarization measurements were performed with a Safire two instrument (Tecan) at 30°C using Corning 384 Well Low Volume Black Round Bottom Polystyrene NBS microplates. Reaction volumes of 20 µL contained a fixed concentration (20 nM) of one of the indicated $^{5-FAM}$CENP-T peptides or of a $^{5-FAM}$CENP-C$^{1-21}$:MIS12C complex. These were mixed with increasing concentrations of the CENP-C$^{1-71}$, CENP-T$^{195-215}$, or MIS12C variants at the indicated concentrations in binding buffer (20 mM Tris-HCl, pH 8, 150 mM NaCl and 1 mM TCEP). The reaction mixtures were allowed to equilibrate for approximately 15 min at room temperature. Fluorescein (5-FAM) was excited with polarized light at 470 nm, and the emitted light was detected at 525 nm through both horizontal and vertical polarizers. No change in the observed signal (or the underlying observed dissociation constant) was detected after one-hour incubation on ice. Polarization values are shown as mean ± standard error of the mean for three replicates. Dissociation constant values (Kd) were obtained by fitting the fluorescence polarization data by non-linear least square method to a single-site binding model using the Origin software.

## Analytical ultracentrifugation

Sedimentation velocity AUC was performed at 42,000 rpm at 20°C in a Beckman XL-A ultracentrifuge. Purified protein samples were diluted to 15–30 µM in a buffer containing 20 mM Tris-HCl, 150 mM NaCl and 1 mM TCEP and loaded into standard double-sector centerpieces. The cells were scanned at 280 nm or 555 nm every minute and 500 scans were recorded for every sample. Data were analyzed using the program SEDFIT (*Schuck, 2000*) with the model of continuous $c(s)$ distribution. The partial specific volumes of the proteins, buffer density and buffer viscosity were estimated using the program SEDNTERP. Data figures were generated using the program GUSSI.

## Low-angle metal shadowing and electron microscopy

Fractions from an analytical size-exclusion chromatography column were diluted 1:1 with spraying buffer (200 mM ammonium acetate and 60% glycerol) and air-sprayed as described (*Baschong and Aebi, 2006*) onto freshly cleaved mica (V1 quality, Plano GmbH) of approximately 2 × 3 mm. Specimens were mounted in a MED020 high-vacuum metal coater (Bal-tec) and dried for ~14 hr. A platinum layer of approximately 1 nm and a 7 nm carbon support layer was subsequently evaporated onto the rotating specimen at angles of 6–7° and 45° respectively. Pt/C replicas were released from the mica on water, captured by freshly glow-discharged 400-mesh Pd/Cu grids (Plano GmbH), and visualized using a LaB$_6$ equipped JEM-400 transmission electron microscope (JEOL) operated at 120 kV. Images were recorded at a nominal magnification of 60,000x on a 4k X 4k CCD camera F416 (TVIPS), resulting in 0.1890 nm per pixel. Particles were manually selected using EMAN2 (*Tang et al., 2007*) and further analyzed using Fiji (*Schindelin et al., 2012*).

## LacO-LacI tethering in U2OS cells

U2OS-lacO cells (gift by G. Kops) (*Janicki et al., 2004*) were maintained in DMEM (PAN Biotech) supplemented with 10% FBS (PAN Biotech), 2 mM L-glutamine (PAN Biotech) and 50 µg/ml Penicillin/Streptomycin (GIBCO). Cells were grown on coverslips pre-coated with 0.01% poly-L-lysine (Sigma-Aldrich) and transfected for 48 hr using a 3:1 ratio of X-tremeGENE (Roche) transfection reagent to pcDNA5/FRT/TO plasmids encoding LacI-EGFP-CENP-T$^{2-373}$ fusion constructs. Nocodazole (Sigma-Aldrich) was added at 330 nM for 14–16 hr before fixation. Cells were permeabilized with a 0.5% triton-X-100 solution in PHEM (Pipes, Hepes, EGTA, MgCl$_2$) buffer for 2 min and fixed with 4% PFA in PHEM for 15 min. After blocking with 3% BSA in PHEM supplemented with 0.1% triton-X-100 (PHEM-T), cells were incubated at room temperature for 1–2 hr with the following primary antibodies diluted in blocking buffer: anti-Nsl1 (in-house generated, mouse monoclonal antibody clone QM 9–13 (27); 1:1000), anti-Hec1 (Gene-Tex GTX70268-100, mouse monoclonal antibody; 1:500), CREST/anticentromere antibody (human autoimmune serum, Antibodies, Inc., 1:100). After washing with PHEM-T, goat anti-human Alexa Fluor 647 (Invitrogen) and goat anti-mouse Rhodamine Red (Jackson Immuno Research) were used as secondary antibodies. DNA was stained with 0.5 µg/ml DAPI (Serva) and Mowiol (Calbiochem) was used as mounting media. Cells were imaged at room temperature using a spinning disk confocal device on a 3i Marianas system equipped with an Axio Observer Z1 microscope (Zeiss), a CSU-X1 confocal scanner unit (Yokogawa Electric Corporation, Tokyo, Japan), 100×/1.4NA Oil Objectives (Zeiss), and an Orca Flash 4.0 sCMOS Camera (Hamamatsu). Images were acquired as z-sections at 0.25 µm, Fluorescence signals were quantified on unmodified 16-bit z-series images using Imaris 7.3.4 32-bit software (Bitplane, Zurich, Switzerland). After background subtraction, the mean NSL1 or NDC80 intensity was normalized to the respective EGFP intensity mean signal. Converted 8-bit maximal intensity projections of representative cells are shown in *Figure 5*.

## Acknowledgements

We are grateful to S Wohlgemuth and I Stender for help with NDC80C and MIS12C purifications and to O Hofnagel and F Müller for assistance with electron microscopy and mass spectrometry. We thank J-M Peters for sharing unpublished reagents and acknowledge J Besler, S van Gerwen, J Keller, K Klare, V Krenn, Y Liu, A Maiolica, S Mosalaganti, D Pan, D Prumbaum, A Schleiffer, and F Villa, as well as all members of the Musacchio laboratory for support, ideas, and discussion. SJ acknowledges the support of a Marie Curie Mobility Fellowship, SEMM's Structured International Post Doc program (SIPOD) and an EMBO long-term fellowship (ALTF 262-2009). AM acknowledges funding by the European Union's 7th Framework Program ERC advanced grant agreement RECEPIANCE and the DFG's Collaborative Research Centre (CRC) 1093. The authors declare no competing financial interests.

# Additional information

## Competing interests

AM: Reviewing editor, *eLife*. The other authors declare that no competing interests exist.

## Funding

| Funder | Grant reference number | Author |
| --- | --- | --- |
| European Research Council | AdG 669686 RECEPIANCE | Andrea Musacchio |
| Deutsche Forschungsgemeinschaft | CRC1093 | Andrea Musacchio |
| European Molecular Biology Organization | ALTF 262-2009 | Sadasivam Jeganathan |

The funders had no role in study design, data collection and interpretation, or the decision to submit the work for publication.

## Author contributions

PH, Conceptualization, Data curation, Validation, Investigation, Visualization, Writing—original draft, Project administration, Writing—review and editing; SJ, Conceptualization, Data curation, Formal analysis, Investigation; AP, Investigation, Methodology; PS, Data curation, Formal analysis, Investigation, Methodology; JJ, Investigation; VK, Data curation, Methodology; FW, Methodology; TB, Data curation, Software, Formal analysis, Investigation, Methodology; AM, Conceptualization, Data curation, Formal analysis, Supervision, Funding acquisition, Investigation, Visualization, Writing—original draft, Project administration, Writing—review and editing

## Author ORCIDs

Pim J Huis in 't Veld, http://orcid.org/0000-0003-0234-6390
Andrea Musacchio, http://orcid.org/0000-0003-2362-8784

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
