## [Decision Letter]

Thank you for submitting your article "Molecular basis of outer kinetochore assembly on CENP-T" for consideration by *eLife*. Your article has been reviewed by three peer reviewers, one of whom, Jon Pines (Reviewer #1), is a member of our Board of Reviewing Editors, and the evaluation has been overseen by Anna Akhmanova as the Senior Editor.

The reviewers have discussed the reviews with one another and the Reviewing Editor has drafted this decision to help you prepare a revised submission.

Summary:

In this study the authors have investigated the binding between the CCAN components CENP-T and CENP-C and the outer kinetochore proteins, Ndc80 and Mis12. The authors used recombinant proteins to demonstrate that CENP-T will bind to Ndc80 and to Mis12-Ndc80 after phosphorylation in vitro using Cyclin B-Cdk1. They identify the phosphorylation sites and show which sites are responsible for binding Ndc80 and Mis12. They use a combination of biochemical and electron microscopy assays to estimate the stoichiometry of the complexes generated and present evidence that CENP-T can bind directly to two molecules of Ndc80 via phospho-threonines 11 and 85, and to one molecule of Mis12 via phospho-serine 201. CENP-C can compete for Mis12 binding to this latter site. Many of the main conclusions have been reported recently in other work. For example, two papers from the Westermann lab (PMIDs: 27524485 and 23334295) and a paper from the Fukagawa lab (PMID: 23334297) previously identified the N-terminal Ndc80 binding peptides in CENP-T and its budding yeast equivalent. Thus, the data in Figure 1 is not new. The clear demonstration of the stoichiometry of binding of two Ndc80 molecules to CENP-T in Figure 2 and Figure 3 is nice, but as these two binding peptides are in distinct regions of the molecule and are each sufficient for Ndc80 binding, this is not a particular surprise. We also note that the recent paper from the Westermann lab showed similar data for the budding yeast CENP-T protein, including similar EM images to show the multiple Ndc80 complexes bound (PMID: 27524485).

The most important new data in this paper relates to the interaction between CENP-T and the Mis12 complex. An interaction with the Mis12 complex had been suggested based on cellular experiments, but a direct interaction has not been shown biochemically. As this interaction appears to be central to building an appropriate outer kinetochore structure on CENP-T, it does have the potential to represent an important new advance.

Essential revisions:

1) It should be investigated whether S201 is actually phosphorylated in vitro. The only evidence provided for this is a list of peptides identified by mass spectrometry with a designation of which phosphorylated residue was "called" by the search algorithm. For the peptide containing the SVQ site, there are three other residues that are identified as phosphorylated in distinct peptides (suggesting only singly phosphorylated peptide were identified). Based on the way that these searches work, there is potential uncertainty in identifying the specific residue.

2) The authors claim that S201 is phosphorylated by Cyclin B-Cdk1 even though the sequence does not conform to the consensus site. In particular S201 is followed by a valine and from the crystal structures of Cdk1 and Cdk2 it is difficult to envisage how this would bind since their active sites have a bulky residue that in almost all substrates requires a proline to for the peptide chain to fit. From the Materials and methods, it appears that the authors used insect cell produced cyclin B-Cdk1 raising the possibility that S201 was phosphorylated by a contaminated kinase. It will be important to eliminate this possibility before publication.

3) If this site is phosphorylated, what proportion of the protein is modified at this site in an in vitro reaction? Based on the presented biochemistry and binding assays, it looks like CDK phosphorylation is fully efficient for promoting the interaction with the Mis12 complex. However, as this would be a non-traditional consensus site for CDK, one would expect its phosphorylation to be substantially less efficient than the other sites in CENP-T. A labeling-based strategy to measure the efficiency of phosphorylation at this site would provide allow the authors to assess the potential relevance of this site.

4) Is this site phosphorylated in vivo? Multiple high throughput proteomics analyses have identified other phosphorylation sites in CENP-T. However, the serine 201 site that the authors focus on has only been identified in a single study, again suggesting that its phosphorylation is more rare (or possibly artefactual). Ideally, this would include a phospho-specific antibody or additional mass spectrometry analysis.

5) If this site is critically relevant, is CENP-T S210A defective in binding to Mis12C and in recruiting Mis12C to kinetochores in human cells? Moreover, what the authors data show is that the serine 201 residue is important for binding to the Mis12 complex. They do not show that the phospho-regulation of this site is key. For example, if they were instead to mutate the neighboring valine or glutamine, this could also block an interaction with the Mis12 complex as they may participate in mediating the interaction. In addition, it would be very useful if the authors could refine the requirements for this binding. All of the binding experiments use an extended portion of CENP-T. If Mis12 complex binding occurs in a limited region (and only requires phosphorylation at the serine 201 residue), they should be able to substantially narrow this binding region.

6) The authors show that CENP-C can disrupt the phospho-CENP-T-Mis12C interaction in vitro. Does CENP-T disrupt CENP-C-Mis12C? The authors should determine the affinity between phospho-CENP-T and Mis12C. If they cannot obtain enough phospho-CENP-T for this measurement, they should attempt to narrow down the Mis12C-binding region to a smaller region and perform the measurements with synthetic phospho-peptides as they did with Ndc80C. If CENP-T binds Mis12C with much weaker affinity than CENP-C, Mis12C will prefer to bind to CENP-C at the kinetochore, making it the major receptor for Mis12C.

[Editors' note: further revisions were requested prior to acceptance, as described below.]

Thank you for resubmitting your work entitled "Molecular basis of outer kinetochore assembly on CENP-T" for further consideration at *eLife*. Your revised article has been favorably evaluated by Anna Akhmanova as the Senior editor and a Reviewing editor.

The manuscript has been substantially improved but there is one remaining issue that needs to be addressed before acceptance:

The phosphorylation of S201 by Cyclin B-Cdk1 is a key point of the manuscript. This does not have a proline at position +1 which, according to the crystal structure of the active site, would be incompatible with binding. The evidence that this is phosphorylated by Cyclin B-Cdk1 relies on baculovirus-produced Cyclin B-Cdk1 that could be contaminated by another kinase. Therefore the authors were asked to use bacterial-expressed Cyclin B-Cdk1 to confirm phosphorylation. In the revised manuscript the authors cite the study by Nori Sagata (Suzuki et al., 2015) that showed phosphorylation at a non-SP consensus motif, to which S201 conforms. The problem with this is that the Suzuki et al. study also used baculovirus-expressed Cyclin B-Cdk1, thus the original problem remains unaddressed. As this is a key point of the manuscript the authors do need to show phosphorylation of S201 by bacterially-produced Cyclin B1-Cdk1.

---

## [Author Response]

*Summary:*

*In this study the authors have investigated the binding between the CCAN components CENP-T and CENP-C and the outer kinetochore proteins, Ndc80 and Mis12. The authors used recombinant proteins to demonstrate that CENP-T will bind to Ndc80 and to Mis12-Ndc80 after phosphorylation in vitro using Cyclin B-Cdk1. They identify the phosphorylation sites and show which sites are responsible for binding Ndc80 and Mis12. They use a combination of biochemical and electron microscopy assays to estimate the stoichiometry of the complexes generated and present evidence that CENP-T can bind directly to two molecules of Ndc80 via phospho-threonines 11 and 85, and to one molecule of Mis12 via phospho-serine 201. CENP-C can compete for Mis12 binding to this latter site. Many of the main conclusions have been reported recently in other work. For example, two papers from the Westermann lab (PMIDs: 27524485 and 23334295) and a paper from the Fukagawa lab (PMID: 23334297) previously identified the N-terminal Ndc80 binding peptides in CENP-T and its budding yeast equivalent. Thus, the data in Figure 1 is not new. The clear demonstration of the stoichiometry of binding of two Ndc80 molecules to CENP-T in Figure 2 and Figure 3 is nice, but as these two binding peptides are in distinct regions of the molecule and are each sufficient for Ndc80 binding, this is not a particular surprise. We also note that the recent paper from the Westermann lab showed similar data for the budding yeast CENP-T protein, including similar EM images to show the multiple Ndc80 complexes bound (PMID: 27524485).*

We agree with the referees that our findings in Figure 1–Figure 3 build partly on recent literature. We also argue, however, that the data in these three figures provide a significantly more detailed overview of the recruitment of NDC80C to phosphorylated CENP-T than it was published in previous work. While the recent work of Stefan Westermann and colleagues (Pekgoz Altunkaya et al., 2016) shows the existence of a stable CENP-HIK:CENP-TW:NDC80C (Ctf3c:Cnn1p:Wip1p:Ndc80c) complex in *S. cerevisiae*, we emphasize that our micrographs of the entire human outer kinetochore capture the CENP-T induced oligomerization of NDC80C in an unequivocal manner (comparing our Figure 3 and Figure 6 with Pekgoz Altunkaya et al., Figure 4).

*The most important new data in this paper relates to the interaction between CENP-T and the Mis12 complex. An interaction with the Mis12 complex had been suggested based on cellular experiments, but a direct interaction has not been shown biochemically. As this interaction appears to be central to building an appropriate outer kinetochore structure on CENP-T, it does have the potential to represent an important new advance.*

To address the reviewers’ main concerns, we have now performed several additional experiments (that resulted in several new figures) to characterize the role of CENP-T as a MIS12 complex binder in more detail. Major additions include:

– A comparison of the sequence surrounding CENP-T S201 with non-canonical CDK1 substrates revealed a conserved (P)xxT/SxxR pattern that was identified as a non- canonical CDK signature in a recent paper (Suzuki et al., Scientific Reports, 2015), which we now cite. The extended sequence alignment is shown in Figure 4.

– Synthetic peptides encompassing residues CENP-T^184-215^, CENP-T^192-215^ and CENP- T^195-215^ all bound to MIS12C in a S201 phospho-dependent manner and competed with CENP-C for the binding to MIS12C. This supports the conclusion that the loss of MIS12C binding by the CENPT^2-373^ S201A mutant represents a direct effect. The binding of the peptides to MIS12C was studied by size-exclusion chromatography (Figure 4) and by fluorescence polarization (Figure 7—figure supplement 1).

– We provide additional details on the identification of the CENP-T peptide containing phosphorylated T195 and/or S201 by mass spectrometry and show that S201 is phosphorylated in CENP-T when the latter is part of the CENP-T:MIS12C complex. This information is included as Figure 4—figure supplement 1.

– We adopted a *lacO*-LacI system to recruit CENP-T constructs to an ectopic chromosome location. This allowed us to test different CENP-T mutants for their ability to recruit MIS12C and NDC80C in vivo. In addition to recapitulating previous findings (Rago et al., Current Biology, 2015), we find that the phosphorylation of CENP-T at position S201 is crucial for the recruitment of MIS12C. These data are shown in Figure 5.

– We now show that CENP- phosphorylated at residue S201 not only forms a complex with MIS12C and NDC80C, but also with KNL1C (KNL1 and ZWINT), thus indicating that P-S201 recruits an entire KMN network. These data are shown in Figure 6—figure supplement 2.

– Our recent structure-function analysis of MIS12C revealed a MIS12C mutant that no longer binds to CENP-C (Petrovic et al., Cell, 2016). This MIS12C mutant does not bind CENP-T either, supporting the idea that CENP-C and CENP-T bind MIS12C with an at least partly overlapping site. In addition, we provide direct evidence that binding of CENP-T and CENP-C to MIS12 is competitive. These data are included in Figure 7.

Below is a point-to-point answer of the referees’ concerns.

*Essential revisions:*

*1) It should be investigated whether S201 is actually phosphorylated in vitro. The only evidence provided for this is a list of peptides identified by mass spectrometry with a designation of which phosphorylated residue was "called" by the search algorithm. For the peptide containing the SVQ site, there are three other residues that are identified as phosphorylated in distinct peptides (suggesting only singly phosphorylated peptide were identified). Based on the way that these searches work, there is potential uncertainty in identifying the specific residue.*

We thank the reviewers for raising this point. The other phosphorylation sites within the relevant peptide that the reviewers refer to are T192 and T195. We do indeed observe a mixture of single, double and triply phosphorylated peptides. However, phosphorylation of T195 and S201 are the predominant post-translational modifications of this peptide. A clarification of the identification of single T195, single

S201 and double T195-S201 phosphorylation events is included as Figure 4—figure supplement 1. This figure also contains two annotated ms/ms spectra to illustrate that individual phospho-sites within this peptide are identified reliably both in vitro and (new information added in the revision, see below) in vivo.

*2) The authors claim that S201 is phosphorylated by Cyclin B-Cdk1 even though the sequence does not conform to the consensus site. In particular S201 is followed by a valine and from the crystal structures of Cdk1 and Cdk2 it is difficult to envisage how this would bind since their active sites have a bulky residue that in almost all substrates requires a proline to for the peptide chain to fit. From the Materials and methods, it appears that the authors used insect cell produced cyclin B-Cdk1 raising the possibility that S201 was phosphorylated by a contaminated kinase. It will be important to eliminate this possibility before publication.*

The reviewer raises an important point that had also puzzled us. In the course of the revision, we realized that the CENP-T sequence encompassing CENP-T S201 contains an evolutionary conserved (P)xxT/SxxR pattern that perfectly matches a signature non- canonical CDK site identified in a nice and systematic recent study (Suzuki et al., Scientific Reports, 2015) (Figure 4). Thus, while it is true that S201 is an anomalous, non S/TP site, it is expected to be a good CDK1 substrate based on this previous study. We thus believe that this site represents a *bona fide* Cdk1 substrate. We have also included the phosphorylation control referred to by the reviewer in Figure 2—figure supplement 1.

*3) If this site is phosphorylated, what proportion of the protein is modified at this site in an* in vitro *reaction? Based on the presented biochemistry and binding assays, it looks like CDK phosphorylation is fully efficient for promoting the interaction with the Mis12 complex. However, as this would be a non-traditional consensus site for CDK, one would expect its phosphorylation to be substantially less efficient than the other sites in CENP-T. A labeling-based strategy to measure the efficiency of phosphorylation at this site would provide allow the authors to assess the potential relevance of this site.*

As already pointed out in the response to the previous point, the site containing S201 is part of a non-canonical CDK1 site. In this revision, we analysed by mass spectrometry the phosphorylation status of CENP-T when it is part of a CENP-T:MIS12C complex. For this, we separated free CENP-T and CENP-T:MIS12C by size-exclusion chromatography and analysed different fractions by mass spectrometry. This revealed that peptides lacking the phosphorylation at residue S201 are present in the input sample and in free CENP-T, but not in the fraction containing CENP-T:MIS12C complex. (Figure 4—figure supplement 1). Thus, our data seem at least consistent with the idea that all the shifted CENP-T molecules (i.e. those binding to MIS12) are modified on S201.

*4) Is this site phosphorylated* in vivo*? Multiple high throughput proteomics analyses have identified other phosphorylation sites in CENP-T. However, the serine 201 site that the authors focus on has only been identified in a single study, again suggesting that its phosphorylation is more rare (or possibly artefactual). Ideally, this would include a phospho-specific antibody or additional mass spectrometry analysis.*

We agree with the reviewer that this is an important point. We pulled-down endogenous CENP-T from HeLa cells and carried out a mass spectrometry analysis. In data summarized and shown in Figure 4—figure supplement 1, we report the identification of phosphorylated S201 or T195/S201 peptides. In Figure 4—figure supplement 1, we also summarize overall evidence, from our study and previous studies, that CENP-T is phosphorylated at positions T195 and S201 in vivo.

*5) If this site is critically relevant, is CENP-T S210A defective in binding to Mis12C and in recruiting Mis12C to kinetochores in human cells? Moreover, what the authors data show is that the serine 201 residue is important for binding to the Mis12 complex. They do not show that the phospho-regulation of this site is key. For example, if they were instead to mutate the neighboring valine or glutamine, this could also block an interaction with the Mis12 complex as they may participate in mediating the interaction. In addition, it would be very useful if the authors could refine the requirements for this binding. All of the binding experiments use an extended portion of CENP-T. If Mis12 complex binding occurs in a limited region (and only requires phosphorylation at the serine 201 residue), they should be able to substantially narrow this binding region.*

We thank the reviewers for bringing up these points. In the original manuscript, we had shown that the phosphorylation of CENP-T^2-373^ is required for the interaction with MIS12C and that a phosphorylated CENP-T^2-373^ S201A mutant does not bind MIS12C. We now show that synthetic peptides encompassing residues CENP-T^184-215^, CENP-T^192-215^ and CENP-T^195-215^ bind to MIS12C in a manner that depends on S201 phosphorylation. We also show that these peptides compete with CENP-C for the binding to MIS12C. The binding of the peptides to MIS12C was studied by size- exclusion chromatography (Figure 4) and by fluorescence polarization (Figure 7 and Figure 7—figure supplement 1).

To address the in vivo relevance of CENP-T phosphorylation at position S201, we adopted a *lacO*-LacI system to tether CENP-T fragments to a non-centromeric region. This strategy enables a direct analysis of the role of CENP-T in MIS12C recruitment that would be harder to perform quantitatively at the endogenous kinetochore, where MIS12 is also recruited by CENP-C. We tethered transiently expressed LacI-GFP-CENP-T^2-373^ fusion proteins to ectopic *lacO* loci and studied the subsequent recruitment of MIS12C

and NDC80C. The results are shown in a separate main figure (Figure 5). Lack of MIS12C recruitment in interphase cells is consistent with the dependency of MIS12C and NDC80C recruitment on CENP-T phosphorylation. Whereas a wild-type CENP-T^2-373^ fragment recruits MIS12C, this recruitment is abolished by the S201A mutation in CENP-T. We also confirmed previously findings obtained using a comparable experimental setup (Rago et al., 2015) showing that the CENP-T T195A mutation, as well as mutations at T11 and T85 that prevent CENP-T:NDC80C interactions, also abolish the recruitment of MIS12C to CENP-T. The basis for this effect of NDC80C-mediated stabilization of MIS12 binding to CENP-T requires further scrutiny.

*6) The authors show that CENP-C can disrupt the phospho-CENP-T-Mis12C interaction* in vitro*. Does CENP-T disrupt CENP-C-Mis12C? The authors should determine the affinity between phospho-CENP-T and Mis12C. If they cannot obtain enough phospho-CENP-T for this measurement, they should attempt to narrow down the Mis12C-binding region to a smaller region and perform the measurements with synthetic phospho-peptides as they did with Ndc80C. If CENP-T binds Mis12C with much weaker affinity than CENP-C, Mis12C will prefer to bind to CENP-C at the kinetochore, making it the major receptor for Mis12C.*

We thank the reviewers for raising this point. As mentioned above, we now performed a MIS12C binding experiment with CENP-T^184-215^, CENP-T^192-215^ and CENP-T^195-215^ peptides and show that the interaction of these peptides with MIS12C depends on the presence of phosphorylated S201. The MIS12C binding affinity of phosphorylated CENP-T^192-215^ was determined by fluorescence polarization measurements. These data are shown in Figure 4, Figure 7, and Figure 7—figure supplement 1.

Our recently described structure-function analysis of MIS12C revealed three aromatic residues in the MIS12 subunit that are important for the orientation of MIS12C’s head1 to the stalk region of MIS12C (Petrovic et al., Cell, 2016). Mutation of these residues prevented binding of CENP-C and also prevented a MIS12C-CENP-T interaction. These data further support that CENP-T and CENP-C bind to MIS12C in a similar fashion and are shown in Figure 7. It would be interesting to figure out if the phosphorylation of CENP-T S201 directly contributes to the MIS12C:CENP-T binding interface, or if it e.g. results in a conformational change in CENP-T that is required for the recruitment of MIS12C. Since the sequence similarities between CENP-C and CENP-T do not go beyond a couple of conserved and clustered positively charged residues, a detailed MIS12C:CENP-T structural analysis would be required to reveal analogies and differences between the binding of CENP-C and CENP-T to MIS12C.

[Editors' note: further revisions were requested prior to acceptance, as described below.]

The manuscript has been substantially improved but there is one remaining issue that needs to be addressed before acceptance:

*The phosphorylation of S201 by Cyclin B-Cdk1 is a key point of the manuscript. This does not have a proline at position +1 which, according to the crystal structure of the active site, would be incompatible with binding. The evidence that this is phosphorylated by Cyclin B-Cdk1 relies on baculovirus-produced Cyclin B-Cdk1 that could be contaminated by another kinase. Therefore the authors were asked to use bacterial-expressed Cyclin B-Cdk1 to confirm phosphorylation. In the revised manuscript the authors cite the study by Nori Sagata (Suzuki et al., 2015) that showed phosphorylation at a non-SP consensus motif, to which S201 conforms. The problem with this is that the Suzuki et al. study also used baculovirus-expressed Cyclin B-Cdk1, thus the original problem remains unaddressed. As this is a key point of the manuscript the authors do need to show phosphorylation of S201 by bacterially-produced Cyclin B1-Cdk1.*

The revised version of the manuscript was reviewed favourably, but the reviewers raised a residual concern related to the question whether Ser201 is a bona fide CDK1:Cyclin B substrate. The same issue had been raised already during the first round of review:

“2) The authors claim that S201 is phosphorylated by Cyclin B-Cdk1 even though the sequence does not conform to the consensus site. In particular S201 is followed by a valine and from the crystal structures of Cdk1 and Cdk2 it is difficult to envisage how this would bind since their active sites have a bulky residue that in almost all substrates requires a proline to for the peptide chain to fit. From the materials and methods it appears that the authors used insect cell produced cyclin B-Cdk1 raising the possibility that S201 was phosphorylated by a contaminating kinase. It will be important to eliminate this possibility before publication.”

In response to this concern, we referred to previous work showing that CDK1:Cyclin B can phosphorylate sequences that do not conform to the S/TP consensus provided other features are present, and we noted that these additional features are indeed present at the CENP-T Ser201 site. We also referred to previous work showing that replacement of Pro with Val is the least damaging substitution for the +1 position of the substrate.

We note that the reviewers’ argument that the preference for Pro at the +1 position is due to a bulky kinase residue is imprecise. The preference for Pro is determined by a very unusual main chain angle of the activation segment, typical of a left-handed helix. This peculiar conformation forces a main chain carbonyl of the activation segment to point away from the position that would allow it to establish a hydrogen bond with the main chain of the substrate, and specifically between the target S/T and the +1 residue. Pro is insensitive to this penalty, because it does not have a main chain amide, and becomes therefore selected through a “negative mechanism”. The implication is that phosphorylation of other sequences is energetically more costly, and not that it cannot take place due to steric hindrance, as implied by the reviewers. It follows that adequate energetic compensation with surrounding sequences might render other substrates more appealing.

In addition to discussing this previous evidence, we had also carried out an experiment showing that the migration shift of CENP-T caused by CDK1:Cyclin B phosphorylation is inhibited in presence of RO-3306, a specific CDK1 inhibitor. Regretfully, this experiment, which admittedly answered the reviewers’ concern only indirectly, had been included in a merged manuscript file that I had uploaded on the *eLife* web site. The editorial office contacted me to let me know that revisions require that individual text and figure files are uploaded and I inadvertently forgot to include this control. I apologize for that. However, in retrospect, I don’t think that the reviewers would have found this sufficient, because when the manuscript was re-reviewed the issue was identified for being central and the new request above was issued.

In response to this, we argue as follows:

First, while we also see that the issue of whether CDK1:Cyclin B is the CENP-T Ser201 kinase in vitro is interesting, we don’t see it as a “key point” of our manuscript, which clearly does not aim to address the issue of CDK1 target specificity in any depth. There are several key points in our manuscript, and certainly key in this context is that Ser201 mediates the interaction of CENP-T with the Mis12 complex when phosphorylated, which we prove beyond reasonable doubt. Key would also be to show that this site is a target of CDK1:Cyclin B in cells. We show that the interaction is suppressed before mitosis, arguing that a mitotic kinase does the job. Inhibiting CDK1 in vivo would not take us very far. So this truly important question will remain unanswered until some proper and complex in vivo sensor can be developed. To summarize, we see the question raised by the reviewers as rather marginal in the context of this manuscript.

Second, the specific request of using bacterially-produced CDK1:Cyclin B1 only appeared after the second review. During the first review, only a generic statement that the issue is given proper attention was raised. As I explained above, we tried to answer this concern by showing that there is substantial prior art. The reviewers did not find our answer sufficient and have now raised a specific suggestion. This may not be an *eLife* style request, in my opinion, unless the deficiency it addressed was at risk of invalidating the paper’s main conclusions or interpretation, which is clearly not the case here. We also note that we do not have this reagent in our laboratory. We are aware of a previously generated expression system – by Tim Hunt’s laboratory – that made use of the frog kinase (we use human reagents throughout). The paper is already gigantic and it reports a very large body of high quality work, none of which requires, for its interpretation, that the kinase is known with absolute accuracy.

Third, even if we failed to reproduce the phosphorylation with bacterial kinase, this would not prove our results with the insect cell kinase wrong, because there is no proof that the bacterially and insect cell expressed kinases behave in exactly the same way (regardless of possible contaminations). On the contrary, if we reproduced it, we would still be ignorant of the kinase that phosphorylates CENP-T at Ser201 in cells.

All this said, we understand that the reviewers put special meaning onto this question. We have therefore added a new figure, Figure 4—figure supplement 4, in which we demonstrate the very high grade of the CDK1:Cyclin B sample we use (panel A). Mass spectrometry analysis of this sample failed to detect any S/T kinases (we have this result in a non-annotated list that I will be very happy to share if you wanted to see it, but this will require annotation, as we had to search a self-compiled insect cell database). All contaminants, with the exception of Hsp70, were detected in trace amounts (a single peptide). We also included the experiment with RO-3306 that I had forgotten to include in the revision (panel B). And finally, we also include a new experiment in which we demonstrate that RO-3306 inhibits the CDK1:Cyclin B mediated interaction of CENP-T with the MIS12 complex (panel C). Furthermore, we have modified the text to state that we cannot exclude the possibility that CENP-T is phosphorylated by a contaminating kinase, but that the evidence we provide makes this possibility rather slim (this statement introduces the description of this new Figure 4—figure supplement 4).